# Allosteric ligands control the activation of a class C GPCR heterodimer by acting at the transmembrane interface

Lei Liu[1,2†], Zhiran Fan[1†], Xavier Rovira[3*], Li Xue[1,2], Salomé Roux[2], Isabelle Brabet[2], Mingxia Xin[1], Jean-Philippe Pin[2*], Philippe Rondard[2*], Jianfeng Liu[1*]

[1]Cellular Signaling Laboratory, International Research Center for Sensory Biology and Technology of MOST, Key Laboratory of Molecular Biophysics of MOE, and College of Life Science and Technology, Huazhong University of Science and Technology, Wuhan, China; [2]Institut de Génomique Fonctionnelle, Université de Montpellier, CNRS, INSERM, Montpellier, France; [3]MCS, Laboratory of Medicinal Chemistry, Institute for Advanced Chemistry of Catalonia (IQAC-CSIC), Barcelona, Spain

**\*For correspondence:**
xavier.rovira@cid.csic.es (XR);
jean-philippe.pin@igf.cnrs.fr
(J-PP);
philippe.rondard@igf.cnrs.fr (PR);
jfliu@mail.hust.edu.cn (JL)

[†]These authors contributed
equally to this work

**Competing interest:** The authors
declare that no competing
interests exist.

**Reviewing Editor:** Andrew C
Kruse, Harvard Medical School,
United States

**Abstract** G protein-coupled receptors (GPCRs) are among the most promising drug targets. They often form homo- and heterodimers with allosteric cross-talk between receptor entities, which contributes to fine-tuning of transmembrane signaling. Specifically controlling the activity of GPCR dimers with ligands is a good approach to clarify their physiological roles and validate them as drug targets. Here, we examined the mode of action of positive allosteric modulators (PAMs) that bind at the interface of the transmembrane domains of the heterodimeric $GABA_B$ receptor. Our site-directed mutagenesis results show that mutations of this interface impact the function of the three PAMs tested. The data support the inference that they act at the active interface between both transmembrane domains, the binding site involving residues of the TM6s of the $GABA_{B1}$ and the $GABA_{B2}$ subunit. Importantly, the agonist activity of these PAMs involves a key region in the central core of the $GABA_{B2}$ transmembrane domain, which also controls the constitutive activity of the $GABA_B$ receptor. This region corresponds to the sodium ion binding site in class A GPCRs that controls the basal state of the receptors. Overall, these data reveal the possibility of developing allosteric compounds able to specifically modulate the activity of GPCR homo- and heterodimers by acting at their transmembrane interface.

## Editor's evaluation

This manuscript builds upon recent structural insights into the $GABA_B$ receptor, an unusual and important member of the G protein-coupled receptor (GPCR) family which functions as an obligate heterodimer. The work investigates positive allosteric modulators (PAMs) of the $GABA_B$ receptor that bind to the heterodimeric interface between transmembrane helix 6 of the two protomers. Through functional characterization of a large panel of mutant receptors, a role for this binding site in conveying agonism by the PAMs is tested. The manuscript also provides evidence for a role of residues deep in the transmembrane domain in regulating both constitutive activity and allosteric agonism in $GABA_B$ receptors. These principles are likely also relevant for other family C GPCRs, suggesting a strategy for drug development targeting this important GPCR family.

**Figure 1.** Allosteric binding sites in the different classes of G protein-coupled receptors (GPCRs). (**A, B**) Scheme and structure of the transmembrane domain (TMD) representative of the diversity of the binding sites for allosteric modulators (filled blue circles or blue spheres) in selected human class A and B GPCRs (muscarinic M2 receptor PDB 4MQT [1], purinergic P2Y1 receptor PDB 4XNV [2], corticotropin-releasing factor receptor 1 PDB 4K5Y [3], and $\beta_2$ adrenergic receptor PDB 5X7D [4]) (**A**), as well as in the class C homodimer mGluR5 (PDB 6N51) bound to a NAM (PDB 4OO9) and the heterodimer GABA$_B$R bound to GS39783 (PDB 6UO8) (**B**). In classes A and B, the allosteric modulators were shown to bind to different sites within and outside of the TM bundle, in contrast to the orthosteric ligand (black triangle).

## Introduction

G protein-coupled receptors (GPCRs) are key players in intercellular communication. They are involved in many physiological functions (*Heng et al., 2013*), and many mutations (*Schoneberg and Liebscher, 2021*) and genetic variants (*Hauser et al., 2018*) of GPCR genes are associated with human diseases. Not surprisingly, GPCRs are major targets for drug development (*Hauser et al., 2017*). Although GPCRs are able to activate G proteins in a monomeric state, they can also form homo- and heteromers (*Ferré et al., 2014*), even in native tissues (*Albizu et al., 2010*; *Rivero-Müller et al., 2010*), often named homo- and heterodimer for simplicity. Such complexes allow allosteric cross-talk between receptors and contribute to a fine-tuning of transmembrane (TM) signaling. Then, modulating the activity of GPCR dimers could offer a new way of controlling physiological functions.

Several approaches were developed to control GPCR dimer activity by targeting the TM domain (TMD) with ligands that could be of potential interest in vivo (*Botta et al., 2020*). One approach being tested was the use of bivalent ligands; that is, two ligands attached by a linker able to bind to each protomer within a dimer (*Huang et al., 2021*). However, each ligand still has the possibility to act on the monomers. Another possibility highly anticipated would be the development of ligands binding at the dimer interface, which would not be able to act on monomers, but instead it would specifically control the dimer activity. Hope for the possible identification of such type of ligands came from the discovery of allosteric modulators binding to sites outside the TM bundle in class A and B GPCRs (*Thal*

*et al., 2018*; *Wang et al., 2021*), in regions possibly involved in receptor dimerization and oligomerization (*Figure 1A*). The first evidence for this type of ligands came from the structure of the class C GABA$_B$ receptor, where two different positive allosteric modulators (PAMs) were reported to bind at the TM interface of this heterodimer made of the two homologous subunits GABA$_{B1}$ (GB1) and GABA$_{B2}$ (GB2) (*Kim et al., 2020*; *Mao et al., 2020*; *Shaye et al., 2020*; *Figure 1B*). No other example has been described yet, including in the other class C GPCRs, such as the metabotropic glutamate (mGlu) receptors where the allosteric modulators that target the TMD bind in the ancestral GPCR cavities within the TMD core (*Doré et al., 2014*; *Wu et al., 2014*; *Figure 1B*).

The GABA$_B$ is activated by γ-aminobutyric acid (GABA), the main inhibitory neurotransmitter in the central system linked to various neurological diseases. This receptor is an attractive drug target for brain diseases (*Bowery, 2006*; *Gassmann and Bettler, 2012*) with therapeutic drugs such as baclofen (Lioresal) and β-phenyl-γ-aminobutyric acid (phenibut) used to treat spasticity (*Chang et al., 2013*), alcohol addiction (*Agabio et al., 2018*), anxiety, and insomnia (*Lapin, 2001*). Auto-antibodies that target GABA$_B$ have previously been identified at the origin of epilepsies and encephalopathies (*Dalmau and Graus, 2018*), and genetic mutations have previously been associated with Rett syndrome and epileptic encephalopathies (*Hamdan et al., 2017*; *Vuillaume et al., 2018*; *Yoo et al., 2017*). GABA$_B$ has a unique allosteric mechanism for signal transduction, in which the binding of an agonist in the extracellular domain of GB1 leads to G protein activation through a rearrangement of the intracellular interface of the TMD of GB2 (*Monnier et al., 2011*; *Shaye et al., 2020*; *Xue et al., 2019*).

PAMs offer a number of advantages over the agonists since they temporally and spatially control enhanced signaling only when the natural ligand is present (*Conn et al., 2014*; *Pin and Prézeau, 2007*; *Schwartz and Holst, 2007*). However, many of these PAMs have an intrinsic agonist activity as they can increase receptor activity in the absence of an orthosteric agonist. These allosteric ligands are called ago-PAMs (*Conn et al., 2014*), in contrast to pure PAMs that have no intrinsic agonist effect. The ago-PAMs could be of tremendous interest for the treatment of patients with genetic variants of GPCRs as more and more are discovered in the exome sequence studies (*Hauser et al., 2018*). Finally, the allosteric agonist activity of these molecules is not predictable yet, and the molecular basis of such activity remains unknown.

In this study, we examined if the commonly available GABA$_B$ PAMs all bind in the GB1-GB2 TM interface and how these PAMs can control the GABA$_B$ heterodimer activity. We show that, indeed, all PAMs bind at the same site, at the active interface of these TMDs, despite their different structures. We also reveal that the agonist activity of these PAMs involves a key region in the central core of the GB2 TMD that also controls the constitutive activity of the receptor. This region is functionally conserved in most GPCRs as it corresponds to the Na$^+$ site found in class A receptors (*Zarzycka et al., 2019*). These data reveal the possibility of developing allosteric compounds able to specifically modulate the activity of GPCR homo- and heterodimers.

## Results

### Different functional properties of the PAMs for GABA$_B$ receptor

We have evaluated the agonist activity and the allosteric modulation of the most studied PAMs (*Urwyler, 2011*) on the recombinant wild-type GABA$_B$ receptors, rac-BHFF (*Malherbe et al., 2008*), CGP7930 (*Urwyler et al., 2001*), and GS39783 (*Urwyler et al., 2003*; *Figure 2A*). These compounds had showed pure PAM or ago-PAM effect on GABA$_B$ activity in different studies both in vitro and in vivo (*Urwyler, 2011*). The binding site of both rac-BHFF (*Kim et al., 2020*; *Mao et al., 2020*) and GS39783 (*Shaye et al., 2020*) in the purified full-length GABA$_B$ receptor was recently reported from structural analyses, including when the receptor is in complex with the G protein (*Shen et al., 2021*). They bind into a pocket formed in the active heterodimer by its TM6s at the interface of the TMDs.

In the present study, we show that rac-BHFF and CGP7930 have intrinsic agonist and PAM activity in different functional assays (*Figure 2B–F*, *Figure 2—figure supplement 1*). In contrast, the agonist activity of GS39783 is weaker, acting more like a pure PAM. These results are in line with our previously reported data (*Lecat-Guillet et al., 2017*). The strong agonist effect of rac-BHFF was revealed by its capacity to activate the GABA$_B$ receptor even in the absence of GABA. It was measured by intracellular calcium release and inositol-phosphate-1 (IP$_1$) accumulation assays in cells coexpressing the chimeric G protein Gα$_{qi9}$ (a Gα$_q$ protein in which the last nine C-terminal residues have been replaced by those

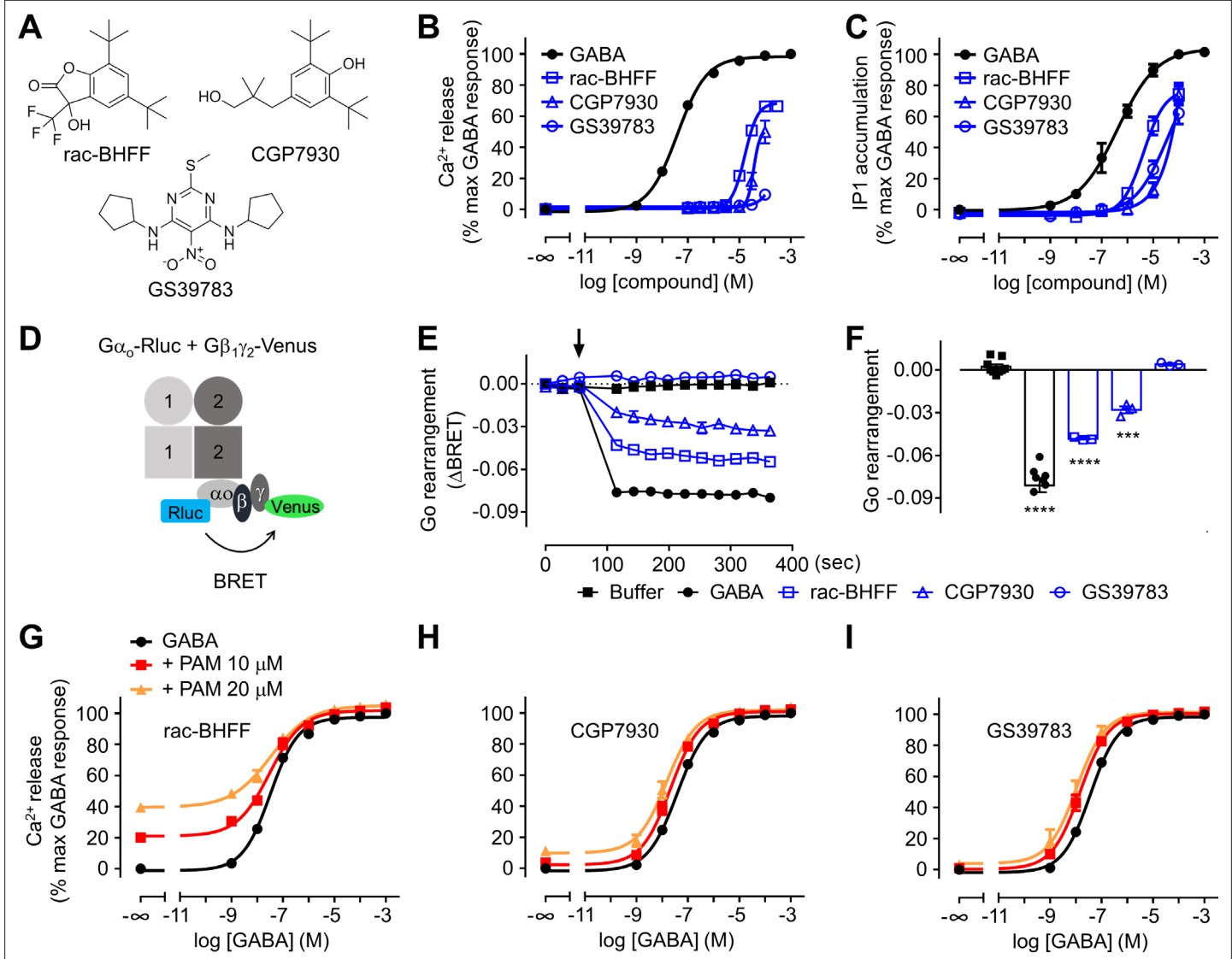

**Figure 2.** Different functional properties of the positive allosteric modulators (PAMs) for GABA$_B$ receptor. (**A**) Chemical structures of the three PAMs of GABA$_B$ receptor used in the study and commercially available. (**B, C**) Intracellular Ca$^{2+}$ responses (**B**) and inositol-phosphate-1 (IP$_1$) accumulation (**C**) mediated by the indicated compounds. (**D**) Schematic representation of the G$_o$ protein BRET sensor. (**E, F**) Kinetics of BRET ratio changes of this sensor (**E**) upon addition (arrow) of buffer (control condition), 100 μM GABA or 100 μM of the indicated PAMs (blue). Data are from a typical experiment performed three times independently. Changes in BRET ratio (**F**) were measured 150 s after drug application. Data are shown as means ± SEM of three biologically independent experiments. Data are analyzed using one-way ANOVA test followed by a Dunnett's multiple comparison test to determine significance (compared with the buffer condition) with ***p<0.0005, ****p<0.0001. (**G–I**) Intracellular Ca$^{2+}$ responses mediated by GABA in the absence or presence of the indicated concentrations of rac-BHFF (**G**), CGP7930 (**H**), or GS39783 (**I**). Data are normalized by the response of 1 mM GABA and shown as means ± SEM of 4–15 biologically independent experiments.

The online version of this article includes the following source data and figure supplement(s) for figure 2:

**Source data 1.** Source data for *Figure 2B–C and F–I*, *Figure 2—figure supplement 1*, and *Supplementary files 1 and 2*.

**Figure supplement 1.** The three positive allosteric modulators (PAMs) show an intrinsic agonist activity on the GABA$_B$ receptor in the inositol-phosphate-1 (IP$_1$) accumulation assay.

of G$\alpha_{i2}$), which allows the coupling of G$_{i/o}$-coupled receptors to phospholipase C (*Figure 2B–F* and *Supplementary files 1 and 2*; *Supplementary file 3*). rac-BHFF alone reached more than 60% of the maximal effect of the full agonist GABA (*Figure 2B and C*, *Figure 2—figure supplement 1A*, and *Supplementary file 1*). CGP7930 also has agonist activity in absence of GABA (*Figure 2B and C* and *Supplementary file 1*). In contrast to the two other PAMs, the intrinsic agonist activity of GS39783

was only observed in the IP$_1$ accumulation assays (*Figure 2C*), and not in the calcium release assay (*Figure 2B* and *Supplementary file 1*), and using a BRET sensor for the activation of Gα$_{oA}$ protein (*Figure 2F*). The observed discrepancy is most likely related to the nature of these assays, the IP$_1$ accumulation assay being an equilibrium assay while the two other are highly dependent on the kinetics of ligand binding to the receptor (*Bdioui et al., 2018*). The IP$_1$ assay is thus proposed to be the most sensitive assay for evaluating slow binders and low-efficacy compounds such as the GS39783.

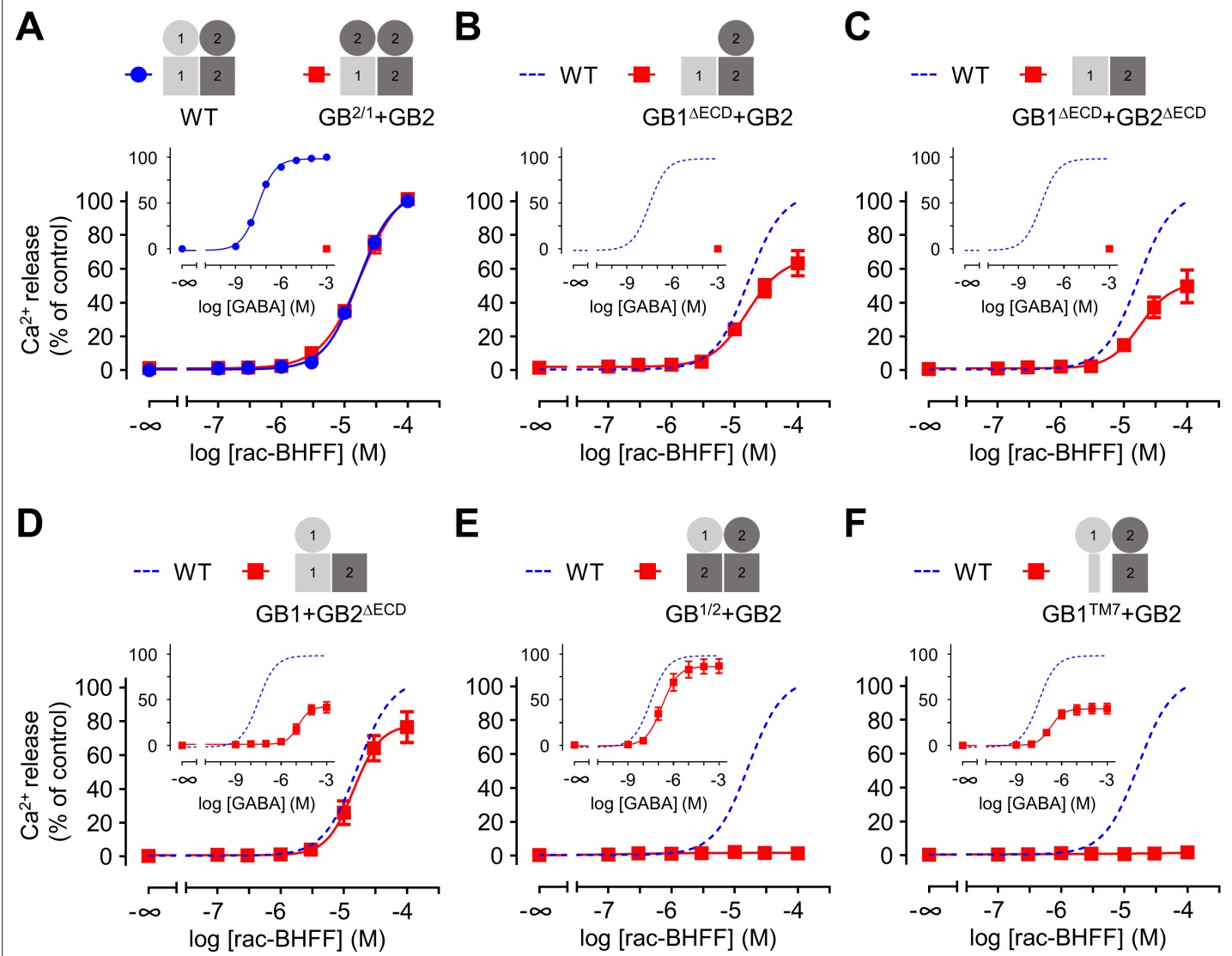

**Figure 3.** Both GB1 and GB2 transmembrane domains (TMDs) are sufficient for the agonist activity of the positive allosteric modulators (PAMs). (**A–F**) Intracellular Ca$^{2+}$ responses mediated by the indicated subunit compositions (pictograms) upon stimulation with rac-BHFF. The inserted graphs correspond to the responses upon stimulation with GABA. Data are normalized by using the response of 100 µM rac-BHFF or 1 mM GABA, for rac-BHFF and GABA treatment, respectively, on wild-type GABA$_B$ receptor and shown as means ± SEM of 3–8 biologically independent experiments. The dotted lines in the main and inserted graphs indicate the dose–responses of the wild-type receptor determined in panel (**A**).

The online version of this article includes the following source data and figure supplement(s) for figure 3:

**Source data 1.** Source data for *Figure 3*, *Figure 3—figure supplement 1*, *Figure 3—figure supplement 2*, and *Supplementary file 3*.

**Figure supplement 1.** GB1 transmembrane domain (TMD) is required for agonist activity of rac-BHFF.

**Figure supplement 2.** GB1 transmembrane domain (TMD) is required for agonist activity of the ago-positive allosteric modulators (ago-PAMs).

## GB1 and GB2 TMDs are sufficient for the agonist activity of the PAMs

Our recent studies have shown that there is strong positive cooperativity between GB1 and GB2 TMDs for receptor activation (*Monnier et al., 2011*; *Xue et al., 2019*). Here, we demonstrate that both GB1 and GB2 TMDs are required for the efficient agonist activity of the PAMs. This is consistent with the binding site of rac-BHFF and GS39783 involving both GB1 and GB2 TMD in the active conformation of the receptor (*Kim et al., 2020*; *Mao et al., 2020*; *Shaye et al., 2020*; *Shen et al., 2021*).

We measured the rac-BHFF-induced intracellular calcium signaling of different GABA_B constructs expressed at the cell surface (*Figure 3*, *Figure 3—figure supplement 1A*, and *Supplementary file 3*). The presence of both GB1 and GB2 TMDs was sufficient for its activity (*Figure 3A–D*). Accordingly, the presence of GB1 or GB2 ECD or both is dispensable for the ago-PAM activity since the relevant constructs in which one or both ECDs were deleted could still be activated by rac-BHFF alone efficiently (*Figure 3B–D*). However, no agonist activity of rac-BHFF could be measured in GABA_B receptors in which the GB1 TMD was replaced by a GB2 TMD or only the seventh helix of the GB1 (GB1-TM7) (*Monnier et al., 2011*; *Figure 3E–F*, *Figure 3—figure supplement 1B and C*), even though these constructs could still be activated by GABA efficiently (*Figure 3E–F*, *Figure 3—figure supplement 1D*). Of note, the conformational state of the GB1 TMD is not critical for both orthosteric and allosteric activation since similar results were obtained with the mutant GB1^DCRC (*Figure 3—figure supplement 1E and F*). This mutant is activated by GABA similarly to the wild-type receptor (*Figure 3—figure supplement 1E*) as previously reported (*Monnier et al., 2011*). But it was engineered to create a disulfide bond in GB1 TMD, thus expecting to limit the conformational change of this domain upon ligand stimulation of the GABA_B receptor. The importance of the intact GB1 TMD in the agonist activity of rac-BHFF was also confirmed by measuring the Go protein activation by BRET (*Figure 3—figure supplement 2A*). Finally, we obtained a similar conclusion regarding the requirement of both GB1 and GB2 TMDs for the ago-PAM activity of the two other PAMs by comparing all of them in the most sensitive IP_1 assays (*Figure 3—figure supplement 2B*). Of note, for CGP7930 there is an apparent controversy between our data where both GB1 and GB2 TMD are required to observe an agonist effect, while *Binet et al., 2004* found GB2 TMD alone was sufficient. It might be due to the endogenous expression of the GB1 subunit that may exist in some cell lines including the HEK293 cells used in both studies (*Xu et al., 2014*).

## GB1 and GB2 TM6s interface is the binding site for the different PAMs

A similar binding site in the structure of the purified GABA_B was reported for rac-BHFF and GS39783 (*Figure 4A*; *Kim et al., 2020*; *Mao et al., 2020*; *Shaye et al., 2020*; *Shen et al., 2021*). This binding site of GS39783 was investigated using receptor mutants expressed at the surface of live cells (*Shaye et al., 2020*), and the mutagenesis data were consistent with the binding site observed in the structure. In the present study, we have analyzed both the potency and agonist efficacy of the three PAMs on a series of both GB1 and GB2 bearing single mutations in their TM6s. First, the potency of the rac-BHFF for each mutant in the intracellular calcium assay was measured in the presence of GABA at a concentration equivalent to its 20% maximal efficacy concentration (EC_{20}). Two single mutants of GB1 (K792^{ICL3}A and Y810^{6.44}A) and three single mutants of GB2 (M694^{6.41}A, Y697^{6.44}A, and N698^{6.45}A) strongly impaired the potency of rac-BHFF for the GABA_B in the calcium assay (*Figure 4B and C*). Residues were named according to the class C GPCR TMD nomenclature (*Isberg et al., 2015*). This loss of activity was not due to a loss of expression of the GABA_B mutants at the cell surface as shown by the cell surface quantification of HALO-tagged GB1 labeled with the fluorophore Lumi4-Tb when coexpressed with GB2 (*Figure 4D*). In addition, the receptor remained functional for all constructs (*Figure 4B and C*, *Figure 4—figure supplement 1A*), although GABA had an impaired activity for the GB1 mutant K792^{ICL3}A (*Figure 4B*) and the GB2 mutants M694^{6.41}A and Y697^{6.44}A (*Figure 4C*). Finally, the critical importance of the two TM6s for the agonist activity of rac-BHFF was confirmed by IP_1 measurements. rac-BHFF had a strong impaired efficacy on two GB1 mutants (Y810^{6.44}A and ^{6.43}MYN^{6.45}-AAA) and one GB2 mutant (^{6.43}MYN^{6.45}-AAA) (*Figure 4E*). The data obtained in the intracellular calcium release assay were also consistent (*Figure 4—figure supplement 2*). This demonstrated that the interface of TM6s is crucial for the agonist activity of rac-BHFF effect. These mutagenesis data are consistent with the binding site of rac-BHFF at the interface between the two GABA_B subunits, similar to GS39783 (*Shaye et al., 2020*), as reported in the GABA_B structures (*Kim et al., 2020*; *Mao et al., 2020*; *Shen et al., 2021*).

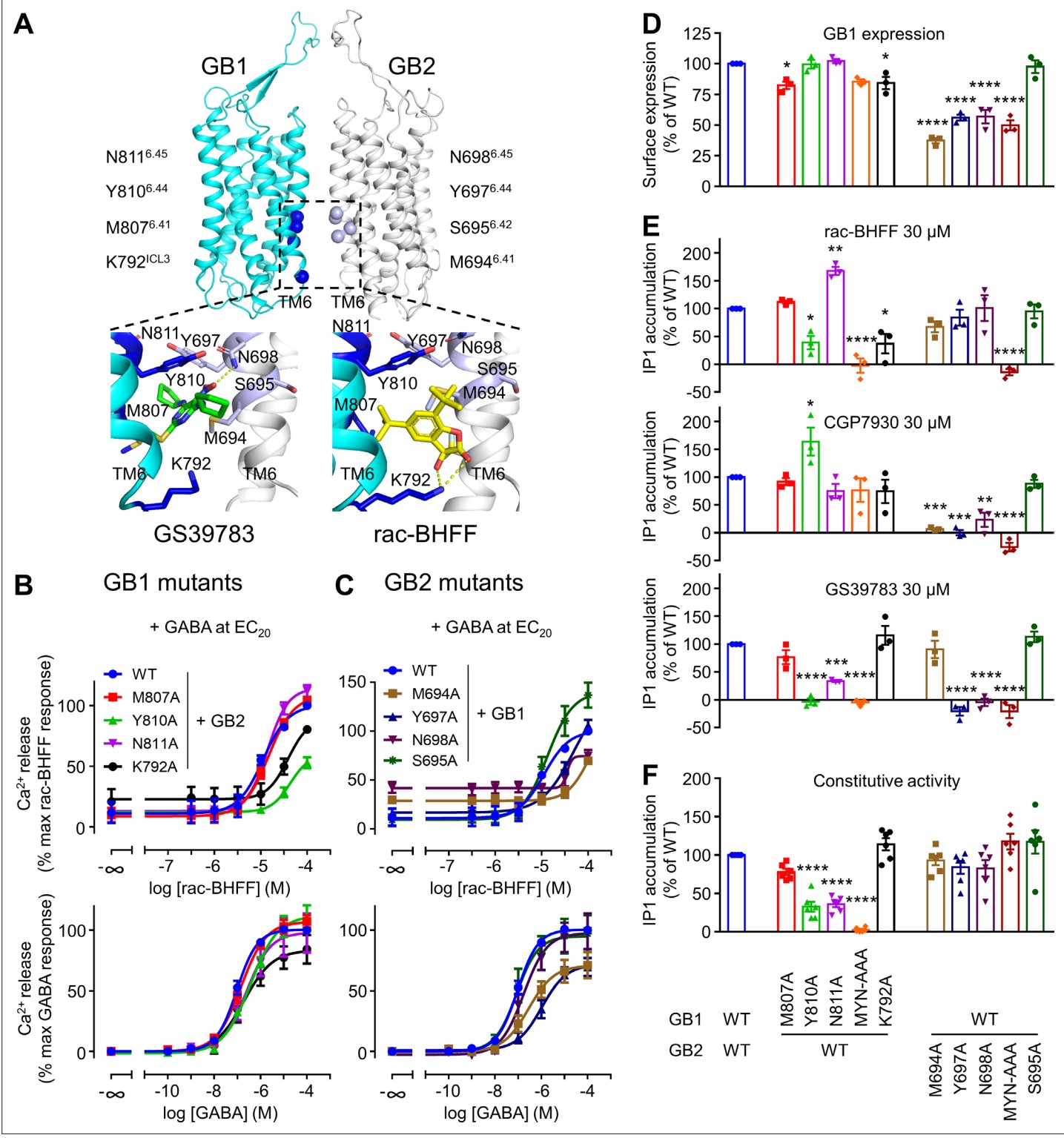

**Figure 4.** GB1 and GB2 TM6s interface is the binding site for the different positive allosteric modulators (PAMs). (**A**) Structure of the GABA$_B$ receptor (PDB 6UO8) where the binding site of GS39783 (PDB 6UO8) and rac-BHFF (PDB 7C7Q) in the receptor is highlighted, and close-up view of the molecules bound (PAMs shown as sticks, and hydrogen bonds between PAMs and receptor are depicted as dashed yellow lines). The α-carbon of the main residues involving in the binding site for these PAMs is highlighted as a sphere in GB1 (blue) and GB2 (light blue). (**B, C**) Intracellular Ca$^{2+}$ responses mediated by the indicated constructs upon stimulation with rac-BHFF in the presence of EC$_{20}$ GABA, or GABA alone. Data are normalized by wild-type response of 100 µM rac-BHFF + EC$_{20}$ GABA or 100 µM GABA, for rac-BHFF and GABA treatment, respectively, and shown as means ± SEM

*Figure 4 continued on next page*

*Figure 4 continued*

of 4–5 biologically independent experiments. (**D**) Quantification of cell surface-expressed GB1 in HEK293 cells transfected with the indicated HALO-tagged GB1 and SNAP-tagged GB2 constructs after labeling with HALO-Lumi4-Tb. Data are normalized by wild-type receptor and expressed as means ± SEM. (**E, F**) Inositol-phosphate-1 ($IP_1$) production induced by the indicated PAMs (**E**) or basal $IP_1$ accumulation (**F**) in intact HEK293 cells expressing the indicated subunit combinations. Data are normalized by wild-type response and shown as means ± SEM of 4–5 biologically independent experiments. Data are analyzed using one-way ANOVA test followed by a Dunnett's multiple comparison test to determine significance (compared with the WT) with *p<0.05, **p<0.005, ***p<0.0005, ****p<0.0001.

The online version of this article includes the following source data and figure supplement(s) for figure 4:

**Source data 1.** Source data for *Figure 4B–F*, *Figure 4—figure supplement 1*, and *Figure 4—figure supplement 2*.

**Figure supplement 1.** GB1 and GB2 TM6s interface is the binding site for the different positive allosteric modulators (PAMs).

**Figure supplement 2.** GB1 and GB2 TM6s interface is the binding site for rac-BHFF.

By using these GB1 and GB2 mutants bearing single mutations in their TM6s, we investigated the previously unknown mode of action of CGP7930. Interestingly, most of the mutations in GB2 strongly impaired the efficacy of CGP7930, whereas such effect was not observed for the mutations in GB1 (*Figure 4E*). Thus, the GB2 TM6 seems more important than the GB1 TM6 for CGP7930 compared to rac-BHFF. Of note, GS39783 is more sensitive to mutations than the two other PAMs. It might be because the mutated residues are highly important for the binding of GS39783 or alternatively to its weakest agonist activity compared to Rac-BHFF and CGP7930, then resulting in a stronger loss of agonist activity of GS39783 on these mutants. Altogether, the three PAMs have a different agonist activity on the GABA$_B$ mutants. Moreover, our mutagenesis data are consistent with the PAMs sharing the same binding pocket at the TM6 TM interface, even though their mode of binding does not seem to involve the same residues in GB1 and GB2.

Finally, we have analyzed the effect of the mutations on the constitutive activity of the GABA$_B$ receptor that was reported in transfected cell lines (*Grünewald et al., 2002*; *Lecat-Guillet et al., 2017*). Indeed, TM6 is expected to control the conformational landscape of the receptor and to play a key role for G protein activation in class A and B GPCRs (*Weis and Kobilka, 2018*). In GABA$_B$ receptor, contacts between the two TM6s were shown to stabilize the active state of the heterodimer (*Kim et al., 2020*; *Mao et al., 2020*; *Shaye et al., 2020*; *Shen et al., 2021*; *Xue et al., 2019*). These contacts are consistent with the allosteric interactions between the GB1 and GB2 TMDs during receptor activation (*Monnier et al., 2011*). In addition, genetic mutations in the GB2 TM6 (S695[6.42]I, I705[6.52]N, and A707[6.54]T) were reported to induce a high constitutive activity of the receptor (*Vuillaume et al., 2018*). Interestingly, in the present study most of the GB1 mutants produced a lower constitutive activity of the receptor compared to the wild-type (*Figure 4F*), for a similar cell surface expression (*Figure 4D*). In contrast, GB2 mutants had a similar constitutive activity as the wild-type receptor, even though the cell surface expression was lower. Of interest, this constitutive activity was blocked by the competitive antagonist CGP54626, on wild-type and most mutated receptors, but it remained unchanged for the triple mutant ([6.43]MYN[6.45]-AAA) in GB2 (*Figure 4—figure supplement 1B*). This suggested that these three mutations in GB2 have impaired the allosteric coupling between the two TMDs in the receptor.

Altogether, these results confirmed the important role of TM6s in controlling the landscape of conformations of the receptor and its basal state. Moreover, it suggests a key role for TM6 allosteric interactions between the GB1 and GB2 TMDs.

## Exploring the GB2 TMD core to clarify the mechanism for agonist activity of the PAMs

How could ago-PAM binding at the active heterodimer interface induce G protein activation by GB2? Binding in the TM6s interface could directly stabilize the active state of the GB2 TMD. Alternatively, and not incompatible with this first mechanism, a second binding site in the GB2 TMD could exist.

A second binding site for the PAM, not yet discovered, could exist in the central core of GB2 TMD as previously proposed (*Dupuis et al., 2006*; *Evenseth et al., 2020*). Binding at this second site per se could be responsible for its agonist activity or alternatively could favor agonist activity of the compound bound in TM6s interface. Possible cavities in GB2 TMD have been revealed by the structures of the receptor, such as those occupied by one phospholipid molecule that covers

nearly the entire range of ligand binding positions previously reported in class A, B, C, and F GPCRs (*Kim et al., 2020*; *Papasergi-Scott et al., 2020*; *Park et al., 2020*). This lipid is present only in the inactive conformation, and it was proposed to be important for intramolecular signal transduction in GABA$_B$ (*Papasergi-Scott et al., 2020*; *Figure 5A*). Mutations designed to destabilize phospholipid binding in GB2 TMD resulted in increases in both GABA$_B$ basal activity and receptor response to GABA. Therefore, to identify a potential second binding site for the PAM, we have introduced mutations in the upper part of the GB2 TMD, the region involved in this phospholipid binding (*Figure 5—figure supplement 1A*). Mutations in this region were shown to confer agonistic activity to GS39783 (*Dupuis et al., 2006*). Single and multiple amino acid substitutions were performed at seven positions in the GB2 (see details in *Figure 5A*, *Figure 5—figure supplement 1A*). We have also changed GB2 L559$^{3.36}$, a residue critical in the ancestral binding pocket of GPCRs, into alanine. It is equivalent to the residue 3.32 in class A GPCRs (Ballesteros–Weinstein numbering scheme [*Ballesteros and Weinstein, 1995*]) that was shown to be conserved for direct interactions with agonists and antagonists (*Venkatakrishnan et al., 2013*). It is also well conserved in the class C GPCRs, including GABA$_B$, where it plays an important role for the NAM and PAM binding and function (*Doré et al., 2014*; *Farinha et al., 2015*; *Feng et al., 2015*; *Leach et al., 2016*; *Wu et al., 2014*).

PAM binding at the TM6s interface might be sufficient to stabilize the active state of GB2 TMD through an allosteric agonist effect. In this scenario, key regions in the GB2 TMD, also called microswitches, should play a key role to stabilize the active state, as well as reported in class A GPCRs (*Zhou et al., 2019*). These microswitches regions remain largely unknown in the class C GPCRs due to the lack of high-resolution structures in the active and inactive states (*Gao et al., 2021*; *Koehl et al., 2019*; *Lin et al., 2021*; *Mao et al., 2020*; *Seven et al., 2021*; *Shaye et al., 2020*; *Shen et al., 2021*). We explored a region underneath the phospholipid cavity, named 'deep region' in this study, that corresponds to the Na$^+$ binding pocket in most class A GPCRs (*Zarzycka et al., 2019*). This region could be reached by synthetic allosteric modulators, as observed by the mGlu5 NAMs that directly interact in this deep region (*Doré et al., 2014*). We changed three residues in GB2 (G526$^{2.46}$, A566$^{3.43}$, and T734$^{7.43}$), where A566$^{3.43}$ interacts directly with the phospholipid (PDB 6WIV) (*Figure 5—figure supplement 1B*), and that are equivalent to the residues that bind Na$^+$ in class A GCPRs (2.50, 3.39, and 7.49) (*Zhou et al., 2019*). Since this Na$^+$ binding pocket collapses during class A GPCR activation, GB2 G526$^{2.46}$ (C$^{2.46}$ in GB1) and A566$^{3.43}$ (G$^{3.43}$ in GB1) were changed for residues larger to fill the cavity (Cys and Phe, respectively). The side chain of T534$^{7.43}$, conserved in GB1, was changed to Ala to prevent these conformational changes during GB2 TMD activation. Single and multiple mutants were analyzed (*Figure 5B*).

## The region of the GB2 TMD equivalent to the sodium binding site in class A GPCRs is critical for allosteric agonism

We have tested the capacity of rac-BHFF to activate the GB2 mutants described above when coexpressed with the wild-type GB1 subunit using the intracellular calcium (*Figure 5B*) and IP$_1$ accumulation (*Figure 5—figure supplement 2A*) assays. rac-BHFF retained its agonist activity for all GABA$_B$ mutants, except for two that contains mutations underneath the lipid binding site, GB2 G526$^{2.46}$C/A566$^{3.43}$F (M14) and G526$^{2.46}$C/T734$^{7.43}$A (M15). The loss of effect of rac-BHFF was not due to the lack of cell surface expression of the mutant receptors (*Figure 5—figure supplement 3A and B*), nor to their loss of function since they were activated by GABA with a similar efficacy as the wild-type receptor (*Figure 5—figure supplement 3C and D*). The agonist activity of CGP7930 was also impaired by these two mutants in the intracellular calcium (*Figure 5—figure supplement 3E*) and IP$_1$ accumulation assays (*Figure 5—figure supplement 2A*). Most importantly, these mutated residues are not critical for the binding of the PAMs. Indeed, rac-BHFF increased intracellular calcium release on the mutants M14 and M15 in the presence of GABA with a potency similar to the wild-type receptor (*Figure 5C*), and improved potency of GABA (*Figure 5—figure supplement 4A*), an effect also observed for CGP7930 and GS39783 (*Supplementary file 2*). Altogether, these results showed that the deep region of the GB2 TMD is required for the agonist activity of the PAMs, but it is not critical for their allosteric modulation effect. It also indicates that this region is not involved in the direct binding of these allosteric compounds.

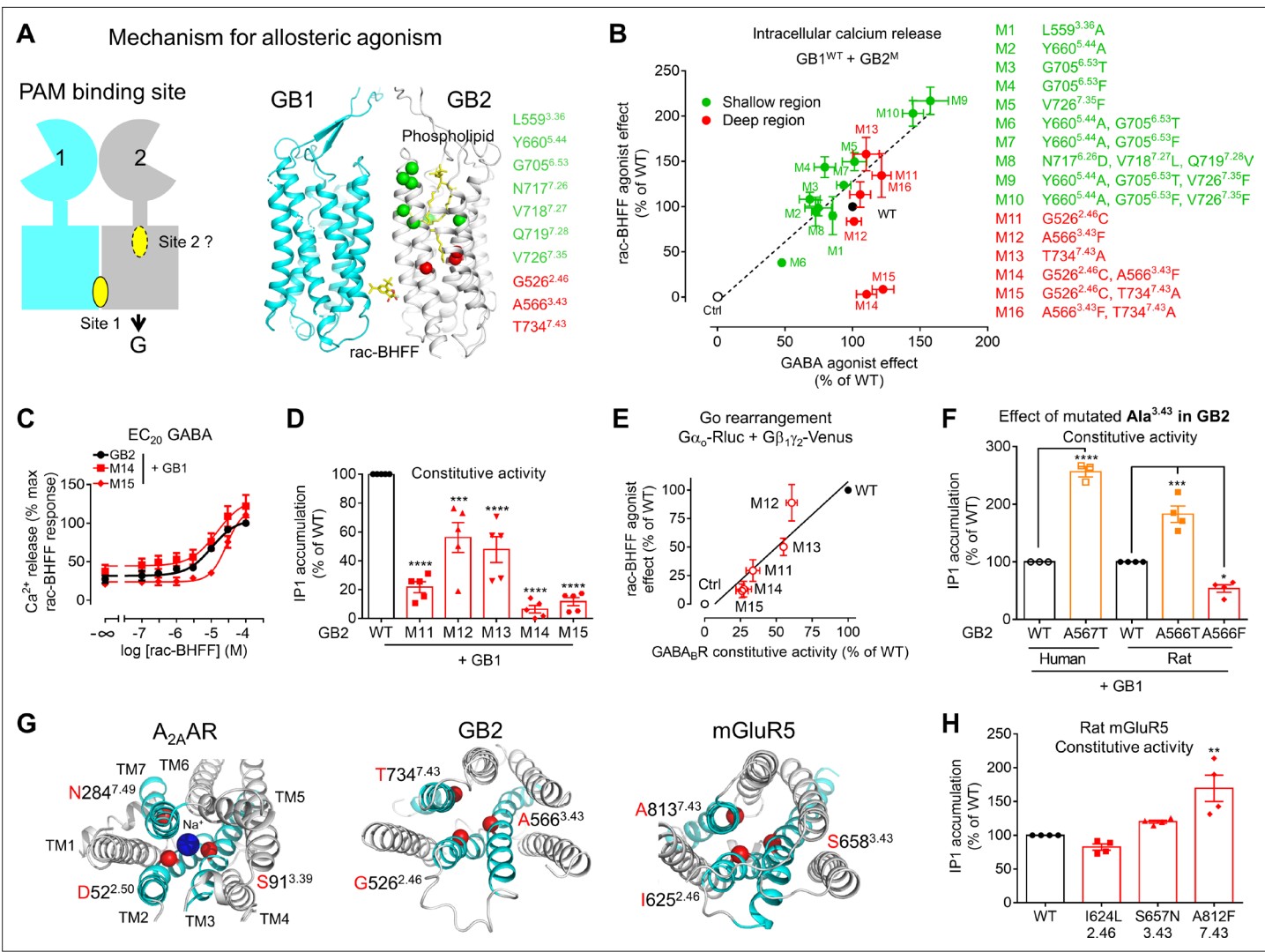

**Figure 5.** A deep region in GB2 transmembrane domain (TMD) is responsible for agonist activity of the positive allosteric modulator (PAM). (**A**) Cartoon highlighting a possible second binding site (dotted oval) for the PAMs in the ancestral ligand binding pocket of the GB2 TMD. In the structure of the inactive state, this pocket is occupied by one molecule of phospholipid (shown as yellow sticks). The residues (α-carbon) of this phospholipid binding pocket (green) and the residues underneath (red) were changed to evaluate their importance in the agonist activity of rac-BHFF. The highly conserved residues L$^{3.36}$ and Y$^{5.44}$ were mutated into Ala; G$^{6.53}$ conserved in GB2 were changed to Thr conserved at this position in GB1, or Phe conserved at this position for other class C GPCRs such as mGlu and CaSR; in Mut 8, the non-conserved residues $^{7.26}$NVQ$^{7.28}$ were mutated in their equivalent in GB2 *Drosophila*; V$^{7.35}$ conserved in GB2 (Val or Phe) was changed to Phe. (**B**) Intracellular Ca$^{2+}$ responses mediated by the indicated GB2 mutants (M1 to M16) coexpressed with the wild-type GB1 subunit upon stimulation with 30 μM rac-BHFF or 1 mM GABA. Data are normalized by wild-type response and expressed as means ± SEM of three biologically independent experiments. (**C**) Intracellular Ca$^{2+}$ responses mediated by the indicated constructs upon stimulation with rac-BHFF in the presence of EC$_{20}$ GABA of each combination. Data are normalized by wild-type response of 100 μM rac-BHFF + EC$_{20}$ GABA and shown as means ± SEM. (**D**) Basal inositol-phosphate-1 (IP$_1$) accumulation mediated by the indicated constructs. Data are normalized by the response of the wild-type and shown as means ± SEM of five biologically independent experiments. Data are analyzed using one-way ANOVA test followed by a Dunnett's multiple comparison test to determine significance (compared with the WT) with ***p<0.0005, ****p<0.0001. (**E**) Correlation between the GABA$_B$ constitutive activity measured using the G$_o$ protein BRET sensor and the rac-BHFF agonist effect for the WT GB1 subunit coexpressed with the indicated GB2 mutants. Data are normalized by the response of the wild-type and shown as means ± SEM. (**F**) Basal IP$_1$ accumulation mediated by the indicated constructs, including the genetic mutation A567T identified in human GB2 that is equivalent to the mutation A566T in rat. Data are normalized by the response of the WT and shown as means ± SEM of 3–4 biologically independent experiments. Data are analyzed using an unpaired *t*-test for human, and one-way ANOVA test followed by a Dunnett's multiple comparison test to determine significance (compared with the WT) for the rat, with *p<0.05, ***p<0.0005, ****p<0.0001. (**G**) Top view of the sodium binding pocket within the structure of human A$_{2A}$ adenosine receptor (PDB 4EIY) where the three residues important for Na$^+$ interactions are highlighted (Cα in red), the equivalent residues identified in human GB2 TMD (PDB 6UO8), and human mGluR5 TMD (PDB 4OO9); the X.50 numbers shown for A$_{2A}$AR are equivalent to the numbers in mGluR5 on the basis of X.50 residues defined in ***Doré et al., 2014***; TM2, TM3, and TM7 that contain the residues involving the identified region are highlighted

*Figure 5 continued on next page*

*Figure 5 continued*

in cyan. (**H**) Basal $IP_1$ accumulation mediated by the indicated WT and mutated mGlu5 receptors in the presence of the co-transfected glutamate transporter EAAT3. Data are normalized by the response of the WT and shown as means ± SEM of four biologically independent experiments. Data are analyzed using one-way ANOVA test followed by a Dunnett's multiple comparison test to determine significance (compared with the WT), with **p<0.005. For clarity, the residue numbers for GB2 subunit are based on the sequence of rat GB2. Negative controls (Ctrl) are HEK293 cells co-transfected with the empty vector and $G\alpha qi_9$ cDNA (**B**), or $G\alpha o$-Rluc and $G\beta_1\gamma_2$-Venus cDNAs in the absence of receptor (**E**).

The online version of this article includes the following source data and figure supplement(s) for figure 5:

**Source data 1.** Source data for *Figure 5B–F and H*, *Figure 5—figure supplement 2*, *Figure 5—figure supplement 3*, *Figure 5—figure supplement 4A and B*, *Figure 5—figure supplement 4E and F*, and *Supplementary file 2*.

**Figure supplement 1.** Phospholipid binding site in the inactive state of GB1 and GB2 transmembrane domain (TMDs).

**Figure supplement 2.** Functional characterization of the GB2 subunit mutants in the transmembrane domain (TMD) core.

**Figure supplement 3.** Characterization of the GB2 subunit mutants in the transmembrane domain (TMD) core.

**Figure supplement 4.** A deep region in GB2 transmembrane domain (TMD) is critical for allosteric agonism.

**Figure supplement 5.** Amino acid sequences alignment of the deep region for both GB1 and GB2.

**Figure supplement 6.** Rearrangements of the deep regions between the inactive and active states of the $GABA_B$ receptor.

## The deep region in the GB2 TMD controls constitutive activity and is involved in human genetic diseases

$GABA_B$ is known to display a significant constitutive activity (*Grünewald et al., 2002*). We have measured this constitutive activity for all the GB2 TMD mutants (*Figure 5—figure supplement 2B*). In contrast to the GB2 mutations in the phospholipid binding site, those in the 'deep region' produced a $GABA_B$ receptor with a constitutive activity strongly impaired compared to the wild-type receptor for a similar cell surface expression (*Figure 5—figure supplement 3A*). The strongest reduction of the constitutive activity was for the GB2 mutants M14 and M15, bearing mutations in the deep region, as measured with both $IP_1$ accumulation (*Figure 5D*, *Figure 5—figure supplement 2B–D*) and BRET (*Figure 5—figure supplement 4B*) assays. It revealed an important role of this region in controlling the basal activity of the $GABA_B$ receptor.

Interestingly, the maximal agonist activity of rac-BHFF correlated with their constitutive activity (*Figure 5E*), demonstrating a possible link between allosteric agonism and constitutive activity. To better understand this relationship, we modeled the agonist activity of PAMs in the $GABA_B$ by developing a mathematical model (*Figure 5—figure supplement 4C*), based on our previously reported mechanistic model for the mGlu receptors (*Rovira et al., 2008*). This previous model was simplified: the extracellular domains were not considered as their effects will be reflected in the basal state of the TMDs. Only one binding site for the allosteric agonist was used and only the GB2 TMD was considered to couple to G proteins. This model coincides then with the two-state model of receptor activation (*Leff, 1995*). It integrates the constitutive activity ($\alpha$) of the $GABA_B$ as well as the binding affinities of the ago-PAM for the active and inactive states of the TMD. According to this model, when the constitutive activity is very low ($\alpha = 100$), the studied PAMs do not efficiently activate the receptor (*Figure 5—figure supplement 4C and D*). This is in agreement with the loss of agonist activity of ago-PAMs on the mutants M14 and M15.

Genetic mutations responsible for human brain diseases such as Rett-like phenotype (*Lopes et al., 2016*; *Vuillaume et al., 2018*; *Yoo et al., 2017*), infantile epileptic spasms (*Hamdan et al., 2017*), and epileptic encephalopathy (*Hamdan et al., 2017*; *Yoo et al., 2017*) were identified in *GABBR2* (which encodes GB2), with most of them resulting in missense mutations in TM3 and TM6. Interestingly, one of the human genetic mutations involved in Rett-like phenotype, GB2 $A567^{3.43}T$, is located in the deep region and also increases receptor constitutive activity. It corresponds to the rat GB2 mutant M12 ($A566^{3.43}F$) that displays a lower constitutive activity (*Figure 5F*). This suggested that depending on the residue at this position the constitutive activity of the receptor can be tuned up or down. This further illustrates the role of the deep region in controlling the conformational landscape of the $GABA_B$.

## The deep GB2 TMD region controls the constitutive activity of other class C GPCRs

The role of this deep region in the controlling of constitutive activity is reminiscent to the role of the equivalent region in class A GPCRs reported to control their conformational landscape (*Manglik*

*et al., 2015*; *Ye et al., 2016*). This is well illustrated by the role of Na$^+$ ions in many class A GPCRs that bind at this topologically equivalent site (*Katritch et al., 2014*; *Ye et al., 2018*; *Zarzycka et al., 2019*; *Figure 5G*). It is also illustrated by the mGlu5 NAMs that interact directly with the same three positions in TM2, TM3, and TM7 in mGluR5 (I$^{2.46}$, S$^{3.43}$, and A$^{7.43}$) (*Doré et al., 2014*), and mGlu4 PAMs (*Rovira et al., 2015*; *Figure 5G*). We have then investigated how mutations of these three residues in mGluR5, equivalent to the residues in GB2 subunits (G$^{2.46}$, A$^{3.43}$, and T$^{7.43}$) but not conserved, can influence its constitutive activity. While two mutations did not change the constitutive activity of mGluR5, the mutation A812$^{7.43}$F increased it (*Figure 5H*, *Figure 5—figure supplement 4E and F*). It suggests that this region is also controlling the conformational landscape of the mGlu5 receptor. But our data show that it is not possible to predict if mutations of these residues in this deep region will produce or not a change in the constitutive activity. Further studies will be necessary to generalize the role of this deep region to the mGlu receptors.

## Discussion

It is a challenging issue to control GPCR dimers' activity with ligands, without acting at the individual monomers. This requires to develop compounds that are not acting at the traditional orthosteric or allosteric binding pockets in the TMD or on the extracellular domain. Here, we used the GABA$_B$ as a model to demonstrate that allosteric modulators can bind in the TMD interface to specifically control GPCR dimer activity. Since these ligands bind to residues that belong to the two protomers, they are unable to act on the individual GABA$_B$ subunits, either GB1 or GB2, that were reported to exist separately (*Chang et al., 2020*). Our study did not identify other binding sites for these PAMs, including those in the GB2 TMD core as it was previously proposed by different groups (*Binet et al., 2004*; *Evenseth et al., 2020*; *Urwyler, 2011*). Our study also reveals that the agonist activity of these PAMs requires a key region in the GB2 TMD core, which is also controlling the constitutive activity of the receptor. This region corresponds to the sodium binding pocket in class A GPCRs.

This extra-TM bundle binding site involving the two TM6s in the GABA$_B$ is novel in the GPCR family. A growing number of allosteric binding sites were discovered outside of the TMD at the receptor-lipid bilayer in class A GPCRs (*Thal et al., 2018*; *Chang et al., 2020*), but TM6 is usually not directly involved, with the exception of class B GPCRs (*Chang et al., 2020*). An important feature to note for this GABA$_B$ TM6s binding pocket for PAMs is that it forms only in the active state. In addition, it is rather tight (volume of 729 Å$^3$) and then it accommodates only small compounds (volumes of 268 Å$^3$ and 260 Å$^3$ for GS39783 and rac-BHFF, respectively). Our study also reveals that the different PAMs have different binding modes. The CGP7930 agonist effect was only impaired by mutations in GB2 TM6 and not by those in the GB1 subunit, suggesting its binding involves mostly GB2. In contrast, rac-BHFF and GS39783 effects were impaired by mutations in GB1 and GB2 TM6s, indicating that both subunits of the GABA$_B$ are involved in the binding of these compounds.

The binding mode of PAMs in the GABA$_B$ is novel within the class C GPCRs where all allosteric compounds that bind in the TMD are found in the ancestral TMD binding pocket, as revealed by the structures of mGlu1 (*Wu et al., 2014*), mGlu2 (*Lin et al., 2021*; *Seven et al., 2021*), mGlu5 (*Doré et al., 2014*), and the calcium sensing receptor CaSR (*Gao et al., 2021*) receptors in complex with allosteric modulators. Similar conclusions were reached for PAMs and NAMs from structure–activity and modeling studies of mGlu (*Bennett et al., 2020*; *Lundström et al., 2017*; *Pérez-Benito et al., 2017*; *Rovira et al., 2015*) and other class C GPCRs (*Leach and Gregory, 2017*; *Leach et al., 2016*). In particular, class C receptors with a deletion of its ECD have shown that PAMs and NAMs bind in the TM bundle, like class A GPCR agonists, and the PAMs behave as agonists on these constructs (*Goudet et al., 2004*). However, in GABA$_B$ the presence of one phospholipid that occupies the ancestral binding pocket both GB1 and GB2 TMD (*Kim et al., 2020*; *Papasergi-Scott et al., 2020*; *Park et al., 2020*) could preclude the possible binding of the PAMs. These phospholipids were proposed to play a role in stabilizing the inactive conformation of the receptor, especially of the GB2 subunit, acting as a NAM (*Papasergi-Scott et al., 2020*; *Park et al., 2020*).

Our study identified a key region where three residues (G$^{2.46}$, A$^{3.43}$, and T$^{7.43}$) are important for the constitutive activity of the GABA$_B$. They are highly conserved during evolution in GB2 subunits (*Figure 5—figure supplement 5*), suggesting that they play a key role in maintaining the resting conformation of the receptor, thereby limiting its basal activity. In mammalian GB1, the residues are slightly different (C$^{2.46}$, G$^{3.43}$, and T$^{7.43}$), but they are also highly conserved. The importance of these

residues in stabilizing the resting state agrees with their direct interaction with the phospholipid ($A^{3.43}$ in GB2 and $T^{7.43}$ in GB1) (*Kim et al., 2020*; *Papasergi-Scott et al., 2020*; *Park et al., 2020*; *Figure 5—figure supplement 1B*). It also corresponds to the equivalent and highly conserved residues $D^{2.50}$, $S^{3.39}$, and $N^{7.49}$ of the class A GPCRs, which are responsible for sodium binding and stabilization of the inactive conformation (*Zarzycka et al., 2019*). But this sodium binding pocket is no longer accessible for the $Na^+$ ions in the active state due to a slight rearrangement of these residues between the active and inactive states (*Zhou et al., 2019*). In GABA_B, the binding of an ion in this region is most probably excluded since most of the residues are hydrophobic. But a significant rearrangement between the inactive and active state in the GB1 and GB2 TMD might occur in this region, even though only slight changes are observed in the available structures (*Figure 5—figure supplement 6*).

The present study also reveals key information to propose a model for the molecular mechanism of activation of the GABA_B (*Figure 6*). First, GB1 TMD plays a key role for the agonist activity of PAMs by providing a binding site at the TM6s interface. In the absence of orthosteric agonist, most probably a fraction of the molecules adopts this TM6s interface state, as observed for the mGlu2 receptor that constantly oscillates between the inactive and active conformation (*Cao et al., 2021*). This TM6s interface provides a binding site for the PAM that further stabilizes this active state. It could explain the agonist activity of the PAM in the absence of another agonist. And in the presence of orthosteric agonist that stabilizes the active TM6 interface (*Mao et al., 2020*; *Shaye et al., 2020*; *Xue et al., 2019*), and also of the coupled G protein (*Shen et al., 2021*), the binding site of the PAM would be favored, then facilitating its agonist and allosteric activities. The PAM would then act as a glue to stabilize the active interface and the active state of the GABA_B receptor. In this simple model, it is difficult to rule out that the PAM induces a new conformation that would not be stabilized by the orthosteric ligands. Finally, the position of both TM6s stabilized by the bound PAM is also consistent with the absence of movement of GB2 TM6 during receptor activation since there is no need to open a cavity for the G protein to bind (*Shaye et al., 2020*; *Shen et al., 2021*). This is a major difference with the other classes of GPCRs (A, B, and F) where a large movement of the intracellular part of TM6 is observed during receptor activation (*Hilger et al., 2020*; *Kozielewicz et al., 2020*). Second, the deep region identified in GB2 TMD serves as a microswitch for ago-PAM to stabilize the active state of GB2 leading to G protein activation. This region plays a key role for both the constitutive activity and allosteric agonism. But one cannot rule out that the mutations in this region of GB2 stabilize the inactive conformation of GABA_B. Finally, since a TM6 active interface is also observed in the mGlu and CaSR dimers (*Gao et al., 2021*; *Koehl et al., 2019*; *Ling et al., 2021*; *Liu et al., 2020*; *Wu et al., 2014*; *Xue et al., 2015*; *Zhang et al., 2020*), it remains to be established if a similar molecular mechanism of activation could be conserved in the mGlu-like class C GPCRs.

In conclusion, our study highlights a distinct mode of action of the PAMs in the GABA_B in a pocket at the TM6 interface that form only in the active state. We demonstrate the importance of the constitutive activity of the GB2 TMD for allosteric agonism through the TMD dimer interface. Our study reveals possibilities of developing novel allosteric modulators for the GABA_B and other class C GPCR dimers through the TM6 interface. Ligands acting at the dimer interface may potentially be interesting tools also for other GPCRs, even if they generally form transient dimers.

## Materials and methods

### Materials

GABA was purchased from Sigma. rac-BHFF, CGP7930, and GS39783 were obtained from Tocris Bioscience. Lipofectamine 2000 and Fluo-4 AM were from Thermo Fisher Scientific. Coelenterazine h was purchased from Promega. Fetal bovine serum (FBS), culture medium, and other solutions used for cell culture were from Thermo Fisher Scientific.

### Plasmids and transfection

The pRK5 plasmids encoding N-terminal HA-tagged wild-type rat GB1a, GB1^ASA, GB1^1/2, GB1^TM7, GB2^2/1, GB1Δ^ECD, and GB1^DCRC, N-terminal Flag-tagged wild-type rat GB2 and GB2Δ^ECD, were described previously (*Monnier et al., 2011*). The mutations of GB2 TMD in the pRK5 plasmid were generated by site-directed mutagenesis using QuikChange mutagenesis protocol (Agilent Technologies). The

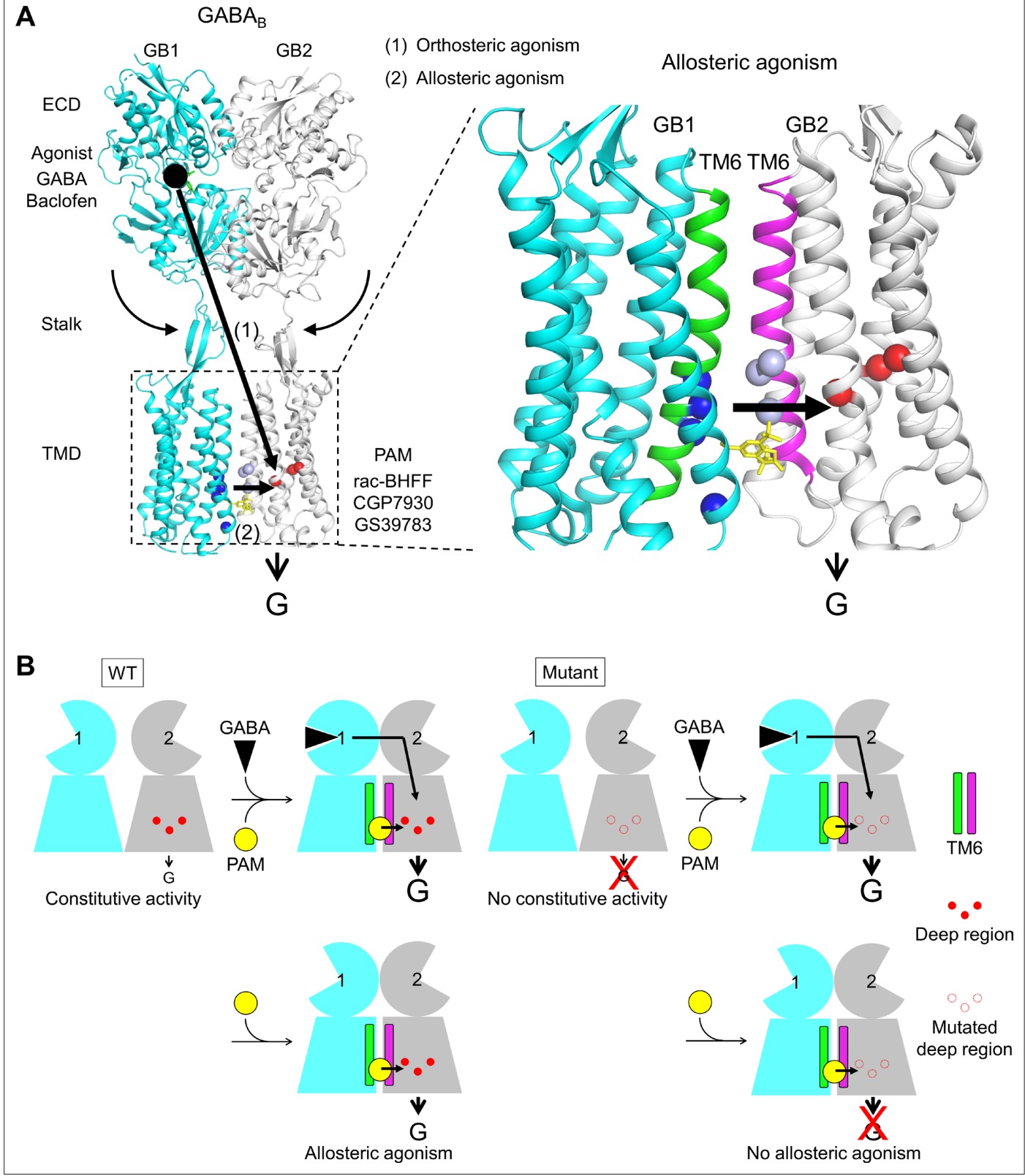

**Figure 6.** Molecular mechanism of GABA$_B$ receptor activation and allosteric modulation. (**A**) The orthosteric agonists bind within the GB1 VFT and induce a rearrangement of ECD dimer. This conformational change stabilizes the active state of GB2 transmembrane domain (TMD) via both the stalk of GB2 and the interactions between the two TMDs through the TM6 dimer interface. This interface in the active state is further stabilized by the positive allosteric modulators (PAMs), then enhancing the potency and affinity of the orthosteric agonists. In the present study, we show that allosteric agonism

*Figure 6 continued on next page*

*Figure 6 continued*

requires a region (residues in red) responsible for basal activity of the receptor. (**B**) This molecular mechanism is further illustrated by pictograms with the WT receptor and with the mutated GB2 TMD deep region. These cartoons further highlight the GB1 TMD that is proposed to serve as a lever for the activation of the receptor both by orthosteric agonists and PAMs.

pRK5 plasmid encoding the N-terminal Flag-tagged wild-type rat mGluR5 was previously described (*Goudet et al., 2004*), and the mutations were generated by site-directed mutagenesis.

HEK293 cells (ATCC, CRL-1573, lot: 3449904) were cultured in Dulbecco's modified Eagle's medium (DMEM) supplemented with 10% FBS. They were tested negative for mycoplasma contamination. They were transfected by electroporation or by the Lipofectamine 2000 protocol as described elsewhere (*Lecat-Guillet et al., 2017*). For different mutations within GB2 TMD, 10 million cells were transfected with 8 µg of each plasmid of interest with 1 µg of wild-type GB1a and completed to a total amount of 10 µg with the plasmid encoding the pRK5 empty vector. For the different combinations of chimeric GABA$_B$ receptor, 2.5 µg of different GB1, 5 µg of different GB2 completed to a total amount of 10 µg with the plasmid encoding the pRK5 empty vector were used as indicated. To allow efficient coupling of the receptor to the phospholipase C pathway, cells were also transfected with the chimeric G-protein Gαq$_{i9}$ (1 µg). For cell-surface expression and functional assays of indicated subunits, experiments were performed after incubation for 36 hr (12 hr at 37°C, 5% CO$_2$ and then 24 hr at 30°C, 5% CO$_2$). For the different mutations within mGluR5 TMD, 10 million cells were co-transfected with 6 µg of wild-type or mutant mGlu5 and 3 µg of EAAT3 (also known as EAAC1) and completed to a total amount of 10 µg with the plasmid encoding the pRK5 empty vector. To eliminate the potential effects of the L-glutamine within classical DMEM, culture medium was replaced by DMEM GlutaMAX 24 hr before experiments.

## Cell surface quantification

Cell surface expression of the indicated subunits was detected by ELISA as previously described (*Liu et al., 2017*). Briefly, HA-tagged different GB1 and Flag-tagged different GB2 were co-transfected into HEK293 cells seeded into 96-well microplates. Cell surface expression and total expression (treated with 0.05% Triton X-100) were detected with the monoclonal rat anti-HA antibody (3F10, Roche) or monoclonal rat anti-Flag (A8592, Sigma) antibody coupled to HRP. Bound antibody was detected by chemiluminescence using SuperSignal substrate (Thermo Fisher Scientific) and a multi-mode microplate reader (FlexStation 3, Molecular Devices). For the mGluR5, Flag-tagged mGlu5 subunit and transporter EAAT3 were co-transfected into HEK293 cells seeded into 96-well microplates.

Cell surface expression of the HALO-tagged GB1 was detected after labeling with the fluorophore Lumi4-Tb, as previously reported (*Scholler et al., 2017*). Briefly, 24 hr after transfection, HEK-293 cells were labeled with 100 nM HALO-Lumi4-Tb in Tag Lite buffer (PerkinElmer Cisbio) for 1 hr at 37°C. After three washes with Tag Lite buffer, the emission of Lumi4-Tb was measured at 620 nm on a PHER-Astar FS microplate reader (BMG LABTECH, Ortenberg, Germany).

## Intracellular calcium release measurements

Intracellular Ca$^{2+}$ release in HEK293 cells was performed as previously described (*Lecat-Guillet et al., 2017*). Briefly, transfected cells in the 96-well plate were washed once with HBSS buffer (20 mM HEPES, 1 mM MgSO$_4$, 3.3 mM Na$_2$CO$_3$, 1.3 mM CaCl$_2$, 0.1% BSA, and 2.5 mM probenecid) and pre-incubated with 1 µM Ca$^{2+}$-sensitive Fluo-4 AM (Thermo Fisher Scientific) prepared in the HBSS buffer for 1 hr at 37°C. Cells were washed once with HBSS buffer and 50 µl of this buffer was added into the wells. The fluorescent signals (excitation at 485 nm and emission at 525 nm) were then measured at intervals of 1.5 s for 60 s after adding of 50 µl of the indicated compounds 20 s after the first reading by the microplate reader (FlexStation 3, Molecular Devices). The Ca$^{2+}$ response was given as the agonist-stimulated fluorescence increase, normalized according to the indication. Concentration–response curves were fitted using 'log(agonist) vs. response -- Variable slope (4 parameters)' by GraphPad Prism software.

## Inositol phosphate measurements

IP$_1$ accumulation in HEK293 cells co-transfected with the indicated subunits was measured by using the 'IP-One Gq assay kits' purchased from PerkinElmer Cisbio according to the manufacturer's

recommendations. In brief, the stimulation buffer with indicated compounds was added into the 96-well microplates and microplates were incubated in the incubator (37°C, 5% $CO_2$) for 30 min. After adding the $IP_1$-d2 and anti-$IP_1$ terbium cryptate conjugate reagents, the microplate was kept in a dark place for 1 hr at room temperature before being detected by the Multi-mode plate reader (PHERAstar FSX, BMG LABTECH). For the mGluR5, Flag-tagged mGlu5 subunit and EAAT3 were co-transfected into HEK293 cells seeded into 96-well microplates.

## BRET signal measurements

The BRET sensor used here is composed of $G\alpha_{oA}$ fused to Rluc ($G\alpha_{oA}$-Rluc), $G\beta_1$ and $G\gamma_2$ fused to Venus ($G\gamma_2$-Venus), which will lead to a BRET signal decreased upon activation of the G protein. BRET measurements were recorded after indicated compounds stimulation on the Mithras LB 940 plate reader (Berthold Biotechnologies, Bad Wildbad, Germany) as previously described (*Comps-Agrar et al., 2011*).

## Binding pocket volume calculations

The calculation of the binding pocket volume was performed with Discovery Studio (BIOVIA, Dassault Systèmes, v20.1.0.19295, San Diego; Dassault Systèmes, 2019). Briefly, the ECD of the full-length $GABA_B$ crystal structure (PDB 6UO8) co-crystallized with GS39783 was truncated and only the TMDs were kept for further analysis. Then, an automatic identification of the binding site was performed from the receptor cavities using the 'Define and Edit Binding Site tools' from the program with the default parameters except the site opening that was set to 8 Å. From all sites identified, the one covering the binding site of PAMs was selected. Then, a structural alignment was performed with the structure co-crystallized with rac-BHFF (PDB 7C7Q). Finally, only those site points at less than 4.5 Å from either GS39783 and rac-BHFF were kept and visually verified. The final volume of the pocket was finally calculated as the product of number of site points and the cube of the grid spacing (0.5 Å).

## Statistical analysis

Data are presented as means ± SEM, and all nonlinear regression analyses of the concentration response curves were performed using GraphPad Prism software. Activation concentration–response curves were fitted to the default equation of 'Log(agonist) vs. response -- Variable slope (4 parameters)' within the software, as $Y = Bottom + (Top-Bottom)/(1 + 10^{((LogEC_{50}-X)*Hill slope)})$ with no constraints for the 4 parameters (X, log of concentration; Y, response; Top and Bottom, plateaus in same units as Y; $LogEC_{50}$, same log units as X; and Hill slope, unit less) in the equation. For activation effects, the 'Top' values in the equation are considered as the 'Emax' for the ligands. Comparison of parameters between different conditions was determined using one-way ANOVA test followed by a Dunnett's multiple comparisons test. For the comparison only containing two members, the unpaired Student's *t*-test was performed. All statistical analyses were performed by GraphPad Prism software, and $p < 0.05$ were considered statistically significant.

## Acknowledgements

We thank Jordi Haubrich for his comments on the manuscript. JL was supported by the Ministry of Science and Technology (grant number 2018YFA0507003), the National Natural Science Foundation of China (NSFC) (grant numbers 81720108031, 81872945, 31721002, and 31420103909), the Program for Introducing Talents of Discipline to the Universities of the Ministry of Education (grant number B08029), and the Mérieux Research Grants Program of the Institut Mérieux. PR and J-PP were supported by the Centre National de la Recherche Scientifique (CNRS, PICS n°07030, PRC n°1403), the Institut National de la Santé et de la Recherche Médicale (INSERM; International Research Program « Brain Signal »), and by grants from the Agence Nationale de la Recherche (ANR-09-PIRI-0011), the FRM (FRM team: DEQ20170326522). XR by the Spanish Ministry of Economy, Industry and Competitiveness (SAF2015-74132-JIN).

# Additional information

## Funding

| Funder | Grant reference number | Author |
|---|---|---|
| Ministry of Science and Technology of the People's Republic of China | 2018YFA0507003 | Jianfeng Liu |
| National Natural Science Foundation of China | 81720108031 | Jianfeng Liu |
| National Natural Science Foundation of China | 81872945 | Jianfeng Liu |
| National Natural Science Foundation of China | 31721002 | Jianfeng Liu |
| National Natural Science Foundation of China | 31420103909 | Jianfeng Liu |
| Ministry of Education of the People's Republic of China | B08029 | Jianfeng Liu |
| Centre National de la Recherche Scientifique | PICS n°07030 | Philippe Rondard |
| Centre National de la Recherche Scientifique | PRC n°1403 | Philippe Rondard |
| Institut National de la Santé et de la Recherche Médicale | IRP Brain Signal | Philippe Rondard |
| Agence Nationale de la Recherche | ANR-09-PIRI-0011 | Philippe Rondard |
| Fondation pour la recherche médicale FRM | FRM team: DEQ20170326522 | Jean-Philippe Pin |
| Ministerio de Economía, Industria y Competitividad, Gobierno de España | SAF2015-74132-JIN | Xavier Rovira |

The funders had no role in study design, data collection and interpretation, or the decision to submit the work for publication.

## Author contributions

Lei Liu, Xavier Rovira, Conceptualization, Data curation, Formal analysis, Writing – original draft; Zhiran Fan, Li Xue, Salomé Roux, Isabelle Brabet, Mingxia Xin, Data curation, Formal analysis; Jean-Philippe Pin, Conceptualization, Formal analysis, Funding acquisition, Writing – original draft, Writing – review and editing; Philippe Rondard, Conceptualization, Formal analysis, Funding acquisition, Supervision, Writing – original draft, Writing – review and editing; Jianfeng Liu, Conceptualization, Formal analysis, Funding acquisition, Supervision, Writing – original draft

## Author ORCIDs

Lei Liu http://orcid.org/0000-0002-9824-9570
Zhiran Fan http://orcid.org/0000-0002-9543-1211
Xavier Rovira http://orcid.org/0000-0002-9764-9927
Salomé Roux http://orcid.org/0000-0002-6106-4863
Jean-Philippe Pin http://orcid.org/0000-0002-1423-345X
Philippe Rondard http://orcid.org/0000-0003-1134-2738
Jianfeng Liu http://orcid.org/0000-0002-0284-8377

## Decision letter and Author response

Decision letter https://doi.org/10.7554/eLife.70188.sa1
Author response https://doi.org/10.7554/eLife.70188.sa2

## Additional files

### Supplementary files

• Supplementary file 1. Allosteric agonist activity of the positive allosteric modulators (PAMs) on the wild-type GABA$_B$ receptor. Data represent the means ± SEM of (n) independent experiments. ****p<0.0001 (one-way ANOVA test); N.A., not applicable; n$_H$, Hill slope.

• Supplementary file 2. Allosteric modulation of the positive allosteric modulators (PAMs) on the indicated GABA$_B$ receptor constructs. Intracellular Ca$^{2+}$ responses mediated by the indicated constructs upon stimulation with GABA in the absence or presence of the indicated concentrations of a PAM. Data represent the means ± SEM of (n) independent experiments. *p<0.05, **p<0.005, ***p<0.0005, ****p<0.0001 (one-way ANOVA test); n$_H$, Hill slope.

• Supplementary file 3. Allosteric agonist activity of rac-BHFF on the indicated 984 GABA$_B$ receptor constructs. Intracellular Ca$^{2+}$ responses mediated by the indicated constructs upon stimulation with rac-BHFF. Data represent the means ± SEM of (n) independent experiments. ***p<0.0005, ****p<0.0001 (one-way ANOVA test); N.A., not applicable; n$_H$, Hill slope.

• Transparent reporting form

### Data availability

Figure 2- Source Data 1 contain the numerical data used to generate the figures; Figure 3 - Source Data 1 contain the numerical data used to generate the figures; Figure 4 - Source Data 1 contain the numerical data used to generate the figures; Figure 5 - Source Data 1 contain the numerical data used to generate the figures.

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
