## [Editor Report]

This manuscript builds upon recent structural insights into the GABA_B_ receptor, an unusual and important member of the G protein-coupled receptor (GPCR) family which functions as an obligate heterodimer. The work investigates positive allosteric modulators (PAMs) of the GABA_B_ receptor that bind to the heterodimeric interface between transmembrane helix 6 of the two protomers. Through functional characterization of a large panel of mutant receptors, a role for this binding site in conveying agonism by the PAMs is tested. The manuscript also provides evidence for a role of residues deep in the transmembrane domain in regulating both constitutive activity and allosteric agonism in GABA_B_ receptors. These principles are likely also relevant for other family C GPCRs, suggesting a strategy for drug development targeting this important GPCR family.

---

## [Decision Letter]

**Decision letter after peer review:**

Thank you for submitting your article "Allosteric ligands control the activation of a GPCR heterodimer by acting at the transmembrane interface" for consideration by *eLife*. Your article has been reviewed by 3 peer reviewers, and the evaluation has been overseen by Andrew Kruse as the Reviewing Editor and Richard Aldrich as the Senior Editor. The following individuals involved in review of your submission have agreed to reveal their identity: Jonathan A Javitch (Reviewer #1); Aashish Manglik (Reviewer #3).

Essential revisions:

1) The reviewers all noted that in a number of places the claims are slightly overstated relative to the underlying data, for example in implying the generalizability of the model to G protein coupled receptors outside of family C. The authors should revise the text to temper their claims and clearly indicate for readers which aspects of the proposed model are direct deductions from experimental results and which aspects are more speculative. Please see detailed reviews below for specific areas to revise.

2) In some places certain expected data are missing, for example the pharmacological characterization is more extensive for rac-BHFF than for CGP7930 or GS39783. Is there a reason that all compounds were not profiled similarly? If so, please explain. If not, then including equivalent data for all three compounds would be preferred.

*Reviewer #1:*

The authors explore the role of an allosteric modulator binding pocket identified in recent structures at the TMD interface of the two GABAB receptor protomers associated with three ago-PAMs and undercover the important role of the TM6s in receptor activation by these PAMs but not by GABA itself. They also identify a key region deep in the GB2 TMD core that is critical not only for the allosteric agonism of these ago-PAMs but also for the constitutive activity of GABAB receptor. Based on these findings, the authors propose a model for the activation of GABAB receptor.

The experiments and their presentation are generally clear and convincing, although there is a tendency in the language to overinterpret the role of mutations as "proving" hypothesis about binding sites rather than supporting the hypothesis, and there is no discussion about a potential role of indirect effects of the mutations. The data are strong and supportive of the inferences drawn, but recognition that they are inferences and not "proof" per se would improve the manuscript further as discussed below in more detail.

By evaluating the agonist activity (ago-PAM) and the positive allosteric modulatory actions of three typical PAMs (rac-BHFF, CGP7930 and GS39783), the authors show both similarities and differences in their functional properties. They show that the GB1 TMD is crucial for the agonist activity of rac-BHFF. Although they also claim that the agonist activity of CGP7930 is also dependent on the GB1 TMD, there are inconsistencies between some of their results and previous literature that need further clarification, as noted below. Through analyzing the potency and agonist efficacy of the three PAMs on a series of both GB1 and GB2 bearing single mutations, the authors provide evidence that the binding sites of rac-BHFF, CGP7930 and GS39783 are at the interface between the two GABAB subunits and that the interface of TM6s is crucial for both their agonist and PAM activity. They use language suggesting that mutations can "prove" that the ligands bind in this position, which should be modified to show that the data support the hypothesis and are consistent with previous crystal structures and at least provide some mention of a caveat that mutagenesis results can of course be confounded by indirect effects and don't prove or even show or demonstrate that this is the binding site but rather support the inference.

The authors also identify a "deep region" within the GB2 TMD core and provide evidence suggesting that it plays a role in stabilizing the active state and is critical for allosteric agonism and constitutive activity. These results are interesting and discussed in the context of the sodium site in Family A receptors. As noted below, there seems to be some selection of the mutants studied here and some additional data would strengthen the argument as discussed in more detail below.

Altogether, this study highlights a distinct mode of action of the PAMs in the GABAB, in a pocket at the TM6 interface that appears to form only in the active state and demonstrates the importance of the GB2-TMD for allosteric agonism through the TMD dimer interface.

Major issues:

1. The authors primarily use two different methods (intracellular ca^2+^ release or IP1 accumulation) to determine the agonist or allosteric modulator activity of three PAMs, and intermittently add BRET data on Go activation. There is one sentence early on about the IP1 assay being more sensitive because of the equilibrium nature of the assay as opposed the kinetic nature of the than ca^2+^ release. The sensitivity of the BRET assay is not discussed. Does the BRET assay reveal agonism for GS39783 as seen in the IP1 assay but not the ca^2+^ release assay? In many figures, the authors only present data from one assay, although sometimes the results are confirmed with another assay. Some description or discussion about the difference or limitation of these two methods and why they are used as they are would be helpful. It is not clear whether the entire data set should be provided in the supplement, but some logic as to why the assays are being used as they are seems important. Furthermore, when loss of function for rac-BHFF is shown by mutations, it would seem prudent to include data for the more sensitive IP1 assay and not only ca^2+^ release.

2. Although the authors claim that they evaluate the agonist activity and the allosteric modulation of all three PAMs, rac-BHFF is the main target of their designed experiments. In some critical results, the data about CGP7930 and GS39783 especially the latter is absent. Since these three PAMs have different functional properties and proposed molecular mechanisms, their data should be presented along with those for rac-BHFF, especially when supporting the conclusion that they have same binding pocket at the TM6s interface.

3. In Figure 3 supplement 2B, the authors show that CGP7930 has no agonist function in the GB1/2-GB2 heterodimer and the GB2 monomer in the IP1 accumulation assay. This conflicts with the data in the paper published in 2004 (Virginie Binet et al., JBC, 2004), which show clearly that CGP7930 can activate GB1/2+GB2 and GB2 monomer without GABA. The authors do not comment upon this discrepancy and how it might be related to the different assay systems – the results seem directly contradictory to this earlier work. Notably the authors select the less sensitive ca^2+^ release assay for their studies of rac-BHFF in this figure. I think it important that these critical data be validated in the IP1 assay as well to establish whether there is any observable agonism in this assay or the direct agonism of the compound is completely gone. Are these issues of assay sensitivity and size of effect?

Additional issues:

1. Figure 1A, the structure panel, is very confusing and the legend does not match the figure. Which structure is shown here or it is a mixture of 4 GPCRs structure? I think it would be helpful to identify the different binding sites somehow so that they can be associated with the specific PDB structures as this is a helpful composite figure if made clearer and with an appropriate legend.

2. In figure 4B and 4C, the curves are very difficult to see. Furthermore, the effects reflect a mixture of agonist and allosteric activity of rac-BHFF. To determine the PAM activity of rac-BHFF, the GABA dose curves in present certain or multiple concentration of rac-BHFF should be presented as in figure 3—figure supplement 1D.

3. Line 156, in Figure 4E, GB1-Y810A and GB1-MYN-AAA only show the TM6s are crucial for the agonist activity of rac-BHFF. This evidence cannot support that GB1 mutants can reduce the PAM activity of rac-BHFF. PAM activity should be removed from the text unless parallel experiments with GABA are shown.

4. Line 162, The data in Figure 4E can only demonstrate CGP7930 effects were impaired by the GB2 mutations but not by GB1 mutations; we cannot make a conclusion that it can activate GB2 alone from these data. This conclusion is also in conflict with the data presented in Figure 3 and its supplemental data where the GB1-TM is shown to be critical even for CGP7930 (Figure 3 S2B).

5. At line 131 there is a conclusion about the GB1DCRC mutant that is locked in an active state still being able to support allosteric activation. Of note, it is also able to support GABA activation. This should be added to text, along with a brief explanation of what it means to be locked in the inactive state if both orthosteric and allosteric activation is intact.

6. Line 152 – there seem to be substantial effects of some of these mutations on GABAB activation, not just the 2 mutations noted.

7. Line 100 – if the lipid is present only in the inactive conformation, then why in the second half of the sentence is it proposed to be important for signal transduction. This should be clarified.

8. Line 244-246 – is it conceivable that the compounds binding in the TMD of GB1 to exert their effects and that they can no longer bind to the deep region in GB2?

9. Line 249 – do any of the green mutations affect constitutive activity? M11, 12 and 13 have effect but emphasis is only on M14 and M15. Would be nice to see all the mutations in this analysis. Are the effects limited to the deep residue mutations or is this more complex?

10. Line 272 – the language is confusing since the different mutations have opposite effects on constitutive activity and this should be clarified.

11. Line 284 -the effect shown in Mglu5 is opposite that shown in GABAB for the same mutation. And two mutations are completely tolerated in the mGlu5 despite their negative effects on GABAB. Caution in generalizing seems prudent here, and I think these are the weakest data in the paper and either should be deleted or strongly caveated. There are many more differences than similarities here.

12. I don't think the concept of GB1 TM domain acting as a lever is a useful or justifiable metaphor. A lever is "a rigid bar resting on a pivot, used to help move a heavy or firmly fixed load with one end when pressure is applied to the other." How agonist binding is conveyed to GB2 and how a PAM affects this is quite complex is terms of the actual physical forces, and I don't believe any of the data here allow this conclusion. Moreover, the authors write "GB1 TMD serves as a lever to stabilize the active state of GMB2 TMD…" As per the definition quoted above, I am not sure how a lever can stabilize. I would remove the lever metaphor altogether. It is also incompatible with the next statement: "This lever model is compatible with the restriction of the TM6s in the active interface where both TM6s do not move substantially between inactive and active states".

This is a matter of taste, but in Figure 2 red and orange are used in B and C for the different PAMS – then in D-F the same colors are used to represent the PAM concentration. I think this is confusing. I would prefer to see consistency in the color schemes in the same and across figures, but this is obviously just my taste.

As best I can tell, Figure 2B red and Figure 2 supplement 1A show the same experimental design, but in Figure 2B a lower range of concentration is shown and the curve doesn't reach the Emax. In the supplemental data, the curve goes to higher concentrations and reaches the Emax, although only two independent experiments are shown. Would have been nice to have a full curve in the main figure. Regardless the legend in the supplemental figure should clarify that it is the same type of experiment as 2B as not clear now why the same experiment is shown twice.

Figure 3 legend should explain the dotted lines.

Figure 5 In panel E I would use the mutant names that are used in all the other panels. I would include all the mutants from panel B in E so that we can see the relationship for both the green and red mutants. Does the relationship hold for all or just the subset shown?

There are a lot of surface expression data. It would be nice to help the reader by adding panels in the supplements showing normalized emax throughout to make it easier for the reader to see the impact of expression – now one can only see that expression isn't dramatically affected but too hard to do it one at a time by mutant.

*Reviewer #2:*

GPCRs are major targets for drug development. A current promising avenue for GPCR drug development is to identify strategies to affect receptor signaling beyond simply targeting the endogenous ligand binding site. This would provide larger surface area to explore and better subtype specificity. Allosteric modulators do this by binding elsewhere in the receptor and enhance or attenuate effect of the orthosteric ligand. The general goal of this research was to investigate the mechanism of such effect in GABAB receptors. Finding the mechanism could inform development of future pharmaceutical drugs. GABAB receptors are members of class C GPCRs. This small family of GPCRs and is somewhat unique in that class C receptors form constitutive dimers. Recent structures of GABAB receptors from 4 different groups showed that 2 GABAB receptor PAMS (rac-BHFF and GS39783) bind at the dimeric interface which is unique among GPCRs. In this study, authors first verified this by performing functional assays and mutagenesis. They also showed the previously described constitutive activity of GABAB receptors and the direct agonist effect of these PAMs (called ago-PAM). Using mutation and structure-function approaches they showed the role of TM6 in both of these properties. Next, they authors tried to use similar assays to investigate the mechanism of agonist activity and modulation by these PAMs. Their results demonstrate the contributions of an internal cavity within the 7-transmembrane domain of GB2, dubbed "deep region", in the constitutive activity of GABAB receptor and the agonist effect of the PAMs but not the allosteric modulations.

While these results provide an interesting twist to how allosteric modulators are perceived and that the pharmacological classification of compounds can be intertwined and not orthogonal, it provides few new mechanistic understanding. For example how does PAMs affect activation (does it modify the energy landscape? or induce a new conformation?) and is it possible to synthesize a PAM that is only a modulator and does not have direct agonist effect? Moreover, the methods section lacks practical details.

Specific comments:

Line 110 – Is it possible the effect of GS39783 in the IP1 assay is due to nonspecific effect? Control titration experiments for figure 2B with cells transfected only with the chimeric G protein or the individual GABAB subunits + the chimeric G proteins would be highly relevant to this data.

Figure 3, figure supplement 1- While it is generally suggested that the ER retention signals inhibit trafficking of GABAB1 subunit to the surface, depending on expression level and cell type this could be leaky. Authors should provide GABAB1 only and GABAB2 only surface expression levels.

Figure 4, panel H. Previous work from authors and others (for example Koehl et al., 2019) has shown that for experiments on mGluR5, co-expression of neuronal excitatory amino acid transporter 3 (EAAT3) is necessary to be able to characterize the effect of modulators correctly. Without that, modulators can appear to show intrinsic agonist activity (for example Koehl et al., 2019). In the data in this panel, EAAT3 was not co-expressed.

Authors have previously shown that GABAB receptors form extensive dimers of dimers and have characterized those. What is the effect of those higher order oligomers on the observed PAM effect?

Line 166 – This explanation for the discrepancy is not clear. The different PAMs tested affect certain mutants in a distinct and sometimes opposite trends (potentiation versus reduction of IP1 accumulation). Specifically, Y810A, MYN-AA, M694A show opposing effects of the different PAMs, while the K792A mutant results in GS39783 showing the greatest potentiating effects.

Line 168 – Based on the data provided (figure 4E) and cited literature (Binet et., 2004; Sun et al., 2016), while the authors showed that CGP7930 rely on key residues in TM6 of GB2 to elicit its agonist effects, this does not imply that these residues are in contact with the compound, forming its binding site versus being a distal component of its allosteric network, or preclude the possibility that CGP7930 binds within or at a different external part of the TMD of GB2. Due to the lack of a structure of GABAB in complex with CGP7930 and the fact that past work shows CGP7930 can directly activate GB2 TMD, the assertion that CGP7930 directly binds at the TM6 interface is under-substantiated. In addition, mutations in the "deep region" within the GB2 TMD pocket, specifically mutants M14 and M15, in principle show that CGP7930 have ablated agonist activity – which can either be caused by the disruption of a "microswitch" necessary for CGP7930 agonism or disruption of a secondary binding site.

Line 190- Kind of related to the previous comment, authors propose 2 general mechanisms for the ago-PAM effect of the compounds, essentially stabilization of the active state via binding of the PAM at the TM6 interface or within a second binding site. The type of experiments they present following up does not directly provide mechanistic insights to verify these models. The mutagenesis work generally reveals involvement of certain residues. This result is not very surprising as these compounds are allosteric and as it has been shown for example with deep mutational scanning on bAR, many mutations within the TMDs affect receptor signaling (Jones et al., *eLife* 2020).

Aside from his argument, these are not the only 2 possible mechanisms for the ago-PAM effect. One very plausible model is that the ago-PAM effect could be due to stabilization of an intermediate state, or even appearance of a new intermediate state upon PAM binding. These reasonable models are not discussed and not included in the mathematical modeling.

Methods section lacks important details for all the assays. For example, Line 397, what concentration of Fluo-4 AM was used? what is the assay buffer? what is the volume? what is the temperature that the assays were performed at?

Line 433 – Fitting equation, parameters, and constraints on the fits should be included. currently it is reported as "Activation- and Inhibition concentration response curves were fitted to the default equations of "log(agonist) vs. response -- Variable slope (four parameters)" and "log(inhibitor) vs. response -- Variable slope (four parameters)" within the software respectively."

*Reviewer #3:*

Liu et al., investigate the mechanism of action of allosteric ligands that affect signaling at the GABAB receptor, a G protein-coupled receptor important in GABA function. The central question is how do these molecules achieve their unique activity, either as direct agonists or as allosteric potentiators of GABA. This study comes at the heels of several cryo-EM structures of GABAB receptors which have revealed the overall architecture of this heterodimeric receptor and the unique between transmembrane domain binding sites for two allosteric modulators that are the focus of this study (rac-BHFF and CGP7930). The authors perform a number of signaling studies to establish that two of these modulators have agonistic and allosteric activity, while one has only allosteric activity. Consistent with the binding site observed in previously determined structures, the authors determine that the transmembrane regions of the GABAB receptor are sufficient for the activity of agonistic positive allosteric modulators. Careful dissection of binding site residues suggests that the interactions made between the three allosteric molecules tested are slightly unique, though their binding sites are likely similar. The authors then test residues deep in the core of the seven transmembrane region of the GABAB subunits, and find that altering this region affects both constitutive activity and the agonistic properties of the modulators, while leaving their allosteric properties intact. With this, the authors propose that the deep site is crucially important for GABAB activation.

While the conclusions specific to GABAB are supported by the authors data, some aspects need to be clarified or revised.

1) It is unclear how well the authors model applies to GPCRs outside of the family C receptors. The authors suggest that the "deep" region is analogous to the sodium binding site of family A GPCRs. These regions differ substantially between the family A and family C GPCRs, and it is unlikely that they serve a similar structural rearrangement upon receptor activation. Furthermore, recent structures of family C receptors bound to G proteins suggest a very unique and divergent mode of G protein engagement compared to family A GPCRs. The authors' speculation around the similarity of activation mechanisms should take these considerations into account.

2) The authors suggest that their findings regarding GABAB modulators that bind at interface between two transmembrane domains likely has implications outside of the family C receptors. The authors should provide appropriate context that, to date, no family A GPCRs have conclusively been shown to form constitutive dimers similar to GABAB or other family C GPCRs.

3) A central new observation in this study is that the deep site is important for constitutive activity and agonistic allosteric modulator activity. The authors propose that this site is a hub for the activation mechanism. Another potential explanation, however, is that the mutations simply stabilize the inactive conformation of the GABAB compared to the wild-type receptor, which has a combined affect on both constitutive activity and the relatively weak agonistic activity of allosteric modulators like rac-BHFF. Indeed, Figure5-supplement 2B shows a rightward shift for GABA dose response for these mutants. The authors should more explicitly discuss potential caveats to their interpretation of the signaling data.

I recommend that the authors revise their speculation that the observations made in this work have general implications for the broader GPCR family outside of family C GPCRs.

---

## [Author Response]

Essential revisions:1) The reviewers all noted that in a number of places the claims are slightly overstated relative to the underlying data, for example in implying the generalizability of the model to G protein coupled receptors outside of family C. The authors should revise the text to temper their claims and clearly indicate for readers which aspects of the proposed model are direct deductions from experimental results and which aspects are more speculative. Please see detailed reviews below for specific areas to revise.

We thanks the reviewers for their constructive comments and justified criticism. In the revised version, we have strongly tuned down the generalizability of our discoveries outside the class C GPCRs, including in the title where we have replaced “GPCR heterodimer” by “class C GPCR heterodimer”. In the revised text, we have clarified the interpretation of the data and the aspects that are more speculative, by using "suggest", "support our hypothesis", or "is consistent with" when it was necessary.

2) In some places certain expected data are missing, for example the pharmacological characterization is more extensive for rac-BHFF than for CGP7930 or GS39783. Is there a reason that all compounds were not profiled similarly? If so, please explain. If not, then including equivalent data for all three compounds would be preferred.

In the revised version, we have now included the equivalent data for the three PAMs. For the wildtype GABA_B_ receptor, the agonist effect of Rac-BHFF, CGP7930 and GS39783 are shown in the three different functional assays, intracellular calcium release (Figure 2B), Go protein rearrangement BRET assay (see new Figure 2D-F) and the most sensitive assay IP1 (Figure 2C and new Figure 2 figure supplement 1). rac-BHFF showed the strongest agonist effect in the three assays, meanwhile GS39783 showed an agonist effect only in the IP1 assay. Now, we have also characterized the agonist effect of the three PAMs for the different GABA_B_ constructs (see new Figure 3—figure supplement 2), and for all the GABA_B_ TMD mutants (Figure 4E and new Figure 5—figure supplement 2) using the IP1 accumulation assay.

Reviewer #1:[…]Major issues:1. The authors primarily use two different methods (intracellular ca^2+^ release or IP1 accumulation) to determine the agonist or allosteric modulator activity of three PAMs, and intermittently add BRET data on Go activation. There is one sentence early on about the IP1 assay being more sensitive because of the equilibrium nature of the assay as opposed the kinetic nature of the than ca^2+^ release. The sensitivity of the BRET assay is not discussed. Does the BRET assay reveal agonism for GS39783 as seen in the IP1 assay but not the ca^2+^ release assay? In many figures, the authors only present data from one assay, although sometimes the results are confirmed with another assay. Some description or discussion about the difference or limitation of these two methods and why they are used as they are would be helpful. It is not clear whether the entire data set should be provided in the supplement, but some logic as to why the assays are being used as they are seems important. Furthermore, when loss of function for rac-BHFF is shown by mutations, it would seem prudent to include data for the more sensitive IP1 assay and not only ca^2+^ release.

We agree with the reviewer. It is important to compare the effect of the three PAMs in the three functional assays. In the revised version, our results with the wild-type GABA-B receptor show that the agonist activity of GS39783 is detected only in the IP-1 assay (Figure 2C and new Figure 2 —figure supplement 1), and not in the BRET assay (see new Figure 2D-F) and in the calcium release (Figure 2B), indicating a higher sensitivity of the IP1 accumulation assay. For the GABA-B constructs, the agonist activity of the three PAMs was compared only in the IP-1 assay (see new Figure 3 —figure supplement 2, panel B), while in the BRET assay only the agonist effect of rac-BHFF could be detected (see new Figure 3 —figure supplement 2, panel A). Altogether it revealed that the IP-1 assay was more sensitive that the BRET and calcium release assays. The lower sensitivity of this calcium and BRET assays is most probably due to the slow binding properties of the PAMs investigated. This feature is much less limiting in the IP-1 assay that relies on the accumulation of IP-1 induced by the PAM for 30 min before measurement.

But for measuring the allosteric modulation of the PAMs, the calcium release assays is enough sensitive. Accordingly, an allosteric modulator activity is observed including for GS39783 (Figure 2I).

Finally, we would like to point out that all the mutants of the GABA-B receptor have been tested in IP-1 assay (see Figure 4E, 5D and new Figure 5 —figure supplement 2, panel A-D). We can thus conclude that the loss of function of the mutants observed, was not due to the poor sensitivity of the assay used.

Accordingly, a better explanation for the sensitivity of the assays has been described in the Results section of the manuscript.

2. Although the authors claim that they evaluate the agonist activity and the allosteric modulation of all three PAMs, rac-BHFF is the main target of their designed experiments. In some critical results, the data about CGP7930 and GS39783 especially the latter is absent. Since these three PAMs have different functional properties and proposed molecular mechanisms, their data should be presented along with those for rac-BHFF, especially when supporting the conclusion that they have same binding pocket at the TM6s interface.

As stated in Point #1 above, the agonist activity of the three PAMs has been now shown for the wild-type receptor in the three functional assays (see Figure 2B-F, and the new Figure 2 —figure supplement 1). We also added the data on the agonist activity of the three PAMs in IP-1 assay for: (i) the different GABA-B constructs (new Figure 3 —figure supplement 2); (ii) the mutants in the TMD interface (Figure 4E); (iii) the mutants in the deep region of GB2 TMD (new Figure 5 —figure supplement 2, panel A, C and D). In addition, the allosteric modulation effect of the three PAMs has been shown for the wild-type receptor by measuring intracellular calcium release (Figure 2G-I).

3. In Figure 3 supplement 2B, the authors show that CGP7930 has no agonist function in the GB1/2-GB2 heterodimer and the GB2 monomer in the IP1 accumulation assay. This conflicts with the data in the paper published in 2004 (Virginie Binet et al., JBC, 2004), which show clearly that CGP7930 can activate GB1/2+GB2 and GB2 monomer without GABA. The authors do not comment upon this discrepancy and how it might be related to the different assay systems – the results seem directly contradictory to this earlier work. Notably the authors select the less sensitive ca^2+^ release assay for their studies of rac-BHFF in this figure. I think it important that these critical data be validated in the IP1 assay as well to establish whether there is any observable agonism in this assay or the direct agonism of the compound is completely gone. Are these issues of assay sensitivity and size of effect?

The reviewer highlights an importance point that has been clarified in the revised version. Indeed there is an apparent controversy between the data reported in Binet et al., (2004) and our current data for CGP7930. We think we did not reproduce the data reported in Binet et al., possibly due to the fact that the cells used in this previous study may have enough endogenous GB1 expression. Indeed, it was shown that the endogenous expression of GB1 in cell lines may vary, including in HEK293 cells (Xu et al., (2014) Angewandte Chemie). This has been now discussed in the Results section and this reference has been added in the revised manuscript.

As discussed above, regarding the sensitivity of the assay used, we now compared the three PAMs in the IP-1 assay (the most sensitive one) for all the GABA-B constructs and mutants to avoid to miss any agonism effect. In the revised version, we thus added the agonist activity of the three PAMs in the IP-1 assay for all the GABA-B constructs (new Figure 3 —figure supplement 2, panel B), including the GB1 and GB2 TMD mutants (Figure 4E and new Figure 5—figure supplement 3, panel A).

Regarding the constructs GB1/2+GB2 and GB2 alone, in contrast to the other constructs, none of the three PAMs were able to induce IP-1 accumulation. It further suggests that the simplest view to explain the difference with Binet et al., is the level of endogenous expression of GB1 in HEK293 cells that was probably higher in this previous study.

Additional issues:1. Figure 1A, the structure panel, is very confusing and the legend does not match the figure. Which structure is shown here or it is a mixture of 4 GPCRs structure? I think it would be helpful to identify the different binding sites somehow so that they can be associated with the specific PDB structures as this is a helpful composite figure if made clearer and with an appropriate legend.

We agree. We added numbers in the Figure 1A to clarify for each allosteric compound its corresponding PDB structure, and we modified the legend accordingly.

2. In figure 4B and 4C, the curves are very difficult to see. Furthermore, the effects reflect a mixture of agonist and allosteric activity of rac-BHFF. To determine the PAM activity of rac-BHFF, the GABA dose curves in present certain or multiple concentration of rac-BHFF should be presented as in figure 3—figure supplement 1D.

To simplify the Figure 4B and 4C we have removed the GB1 and GB2 triple mutants in the revised figure. Indeed, since Rac-BHFF and GS39783 have no agonist activity on these triple mutants in the IP-1 assay (Figure 4E), it is not useful to measure their potency in calcium release assay.

As suggested by the reviewer, for all the mutants at the TMD interface we added the allosteric modulator activity of rac-BHFF on the GABA dose curves in presence of two different concentrations of rac-BHFF in the calcium release assay (see new Figure 4—figure supplement 2). Interestingly, these results revealed a loss of agonist activity of rac-BHFF for all GABA-B mutants, except GB1-M807A. It is in contrast to the lower reduction of this agonist activity measured in the IP-1 assay for all these TMD interface mutants (Figure 4E). This is consistent again with a higher sensitivity of the IP-1 assay.

These data showed that the residue M807 in GB1 plays only a minor role in rac-BHFF activity, as observed in Figure 4B (Rac-BHFF has a similar agonist potency for the WT and mutant GB1-M807A) and Figure 4E (Rac-BHFF has a similar agonist efficacy for the WT and mutant GB1-M807A). This residue M807 seems also no critical for the agonist activity of the two others PAMs (Figure 4E).

3. Line 156, in Figure 4E, GB1-Y810A and GB1-MYN-AAA only show the TM6s are crucial for the agonist activity of rac-BHFF. This evidence cannot support that GB1 mutants can reduce the PAM activity of rac-BHFF. PAM activity should be removed from the text unless parallel experiments with GABA are shown.

We agree for this misunderstanding. We have removed the PAM activity in this sentence.

4. Line 162, The data in Figure 4E can only demonstrate CGP7930 effects were impaired by the GB2 mutations but not by GB1 mutations; we cannot make a conclusion that it can activate GB2 alone from these data. This conclusion is also in conflict with the data presented in Figure 3 and its supplemental data where the GB1-TM is shown to be critical even for CGP7930 (Figure 3 S2B).

We agree with the reviewer’s comments. We have removed the last part of the sentence (Line 163164; “in agreement with CGP7970 being able to active GB2 alone as previously reported”).

5. At line 131 there is a conclusion about the GB1DCRC mutant that is locked in an active state still being able to support allosteric activation. Of note, it is also able to support GABA activation. This should be added to text, along with a brief explanation of what it means to be locked in the inactive state if both orthosteric and allosteric activation is intact.

We agree with the reviewer’s comments that it requires a clarification. We modified the sentence as follow:

“…the conformational state of the GB1 TMD is not critical for both orthosteric and allosteric activation since similar results were obtained with the mutant GB1^DCRC^ (*Figure 3—figure supplement 1E-F*). This mutant is activated by GABA similarly to the wild-type receptor (*Figure 3figure supplement 1E*) as previously reported (Monnier et al., 2011). But it was engineered to create a disulphide bond in GB1 TMD thus expecting to limit the conformational change of this domain upon ligand stimulation of the GABAB receptor.”

6. Line 152 – there seem to be substantial effects of some of these mutations on GABAB activation, not just the 2 mutations noted.

We agree. We have included the mutant GB1-K792A in this sentence. We did not include the GB2 triple mutant since we have removed it from Figure 4C (see Point #2 above).

7. Line 100 – if the lipid is present only in the inactive conformation, then why in the second half of the sentence is it proposed to be important for signal transduction. This should be clarified.

We think the reviewer refers to Line 201 (and not 100). Skiniotis’s group was the only one to investigate the possible importance of this lipid for the functioning of the receptor. Mutations designed to destabilize phospholipid binding in GB2 TMD resulted in the increase in both GABA-B basal activity and response to GABA (Papasergi-Scott et al., (2020) Nature). One sentence has been added after Line 201 to clarify this point in the revised manuscript.

8. Line 244-246 – is it conceivable that the compounds binding in the TMD of GB1 to exert their effects and that they can no longer bind to the deep region in GB2?

Currently, it is not possible to exclude that these PAMs bind in the TMD of GB1. But we don’t think so. As scientists, we tend to always favor the simplest explanation (binding of these PAMs at the TMD heterodimer interface, as indicated by our results), but we must be open to alternative.

9. Line 249 – do any of the green mutations affect constitutive activity? M11, 12 and 13 have effect but emphasis is only on M14 and M15. Would be nice to see all the mutations in this analysis. Are the effects limited to the deep residue mutations or is this more complex?

We agree with the reviewer. To give a better picture of what’s happen in the GB2 TMD, we now provide the results for the constitutive activity of all mutants of GB2 (from M1 to M16; see new Figure 5—figure supplement 2B). While all the mutants in the “deep region” (“red mutations”) have a strongly impaired or abolished constitutive activity, many mutants in the lipid binding pocket (“green mutations”) have a similar constitutive activity that the WT receptor. Only the green mutant M9 has no constitutive activity while it is normally expressed at the cell surface (see ELISA for GB1 and GB2-M9 in the Figure 5—figure supplement 3A). It suggested that the three mutations in GB2-M9 have strongly impaired the ability of the GABAB receptor to adopt an active state in the basal conditions, or alternatively, they have stabilized the inactive state of the receptor. Further studies will be necessary to clarify this effect. In contrast, the low constitutive activity of the mutants M8 and M10 might be due, at least partially, to the lower expression of the mutated receptors (see ELISA for GB1 and GB2-M9 in the Figure 5—figure supplement 3A).

10. Line 272 – the language is confusing since the different mutations have opposite effects on constitutive activity and this should be clarified.

We agree with the reviewer. We have clarified the text of this paragraph in the revised version as follow:

“Interestingly, one of the human genetic mutations involved in Rett-like phenotype, GB2 A567^3.43^T, is located in the deep region and also increases receptor constitutive activity. It corresponds to the rat GB2 mutant M12 (A566^3.43^F) that displays a lower constitutive activity (Figure 5F). This suggested that depending of the residue at this position, the constitutive activity of the receptor can be tuned up or down. This further illustrates the role of the deep region in controlling the conformational landscape of the GABAB.”

11. Line 284 -the effect shown in Mglu5 is opposite that shown in GABAB for the same mutation. And two mutations are completely tolerated in the mGlu5 despite their negative effects on GABAB. Caution in generalizing seems prudent here, and I think these are the weakest data in the paper and either should be deleted or strongly caveated. There are many more differences than similarities here.

We agree with the reviewer. We have clarified the text of this paragraph in the revised version and we avoid to generalize the results obtained with GABA-B to the mGluRs. The text on the mGluR5 results is now as follow:

“We have then investigated how mutations of these three residues in mGluR5, equivalent to the residues in GB2 subunits (G^2.46^, A^3.43^ and T^7.43^) but not conserved, can influence its constitutive activity. While two mutations did not change the constitutive activity of mGluR5, the mutation A812^7.43^F increased it (Figure 5H, Figure 5—figure supplement 4E-F). It suggests that this region is also controlling the conformational landscape of the mGlu5 receptor. But our data show that is not possible to predict if mutations of these residues in this deep region will produce or not a change in the constitutive activity. Further studies will be necessary to generalize the role of this deep region in the mGlu receptors.”

12. I don't think the concept of GB1 TM domain acting as a lever is a useful or justifiable metaphor. A lever is "a rigid bar resting on a pivot, used to help move a heavy or firmly fixed load with one end when pressure is applied to the other." How agonist binding is conveyed to GB2 and how a PAM affects this is quite complex is terms of the actual physical forces, and I don't believe any of the data here allow this conclusion. Moreover, the authors write "GB1 TMD serves as a lever to stabilize the active state of GMB2 TMD…" As per the definition quoted above, I am not sure how a lever can stabilize. I would remove the lever metaphor altogether. It is also incompatible with the next statement: "This lever model is compatible with the restriction of the TM6s in the active interface where both TM6s do not move substantially between inactive and active states".

We agree with the arguments of the reviewer. Accordingly, we have removed the lever metaphor, and we have rephrased the corresponding paragraph in the Discussion.

This is a matter of taste, but in Figure 2 red and orange are used in B and C for the different PAMS – then in D-F the same colors are used to represent the PAM concentration. I think this is confusing. I would prefer to see consistency in the color schemes in the same and across figures, but this is obviously just my taste.

We agree. We have changed the colors used in Figure 2B and C, and in the new Figure 2E and 2F. Now, red and orange are used only to represent the PAM concentrations (see Figure 2G-I).

As best I can tell, Figure 2B red and Figure 2 supplement 1A show the same experimental design, but in Figure 2B a lower range of concentration is shown and the curve doesn't reach the Emax. In the supplemental data, the curve goes to higher concentrations and reaches the Emax, although only two independent experiments are shown. Would have been nice to have a full curve in the main figure. Regardless the legend in the supplemental figure should clarify that it is the same type of experiment as 2B as not clear now why the same experiment is shown twice.

We agree. We have transferred all the data of this former Figure 2-suppplement 1 in the main figure (see new Figure 2) of the revised manuscript.

Figure 3 legend should explain the dotted lines.

The dotted lines in the main and inserted graphes indicate the dose-responses of the wild-type receptor determined in panel *A*. It is now clearly indicated in the revised version.

Figure 5 In panel E I would use the mutant names that are used in all the other panels. I would include all the mutants from panel B in E so that we can see the relationship for both the green and red mutants. Does the relationship hold for all or just the subset shown?

We agree. In panel E, we used the mutant names that are used in all the other panels of Figure 5. We have also performed additional IP1 assays to show that the agonist activity of the three PAMs is correlated with the constitutive activity of the GABAB receptor for the mutants of the “deep region”, but also in some extent for those of the phospholipid binding pocket (see new Figure 5- Supplement 2C-D).

There are a lot of surface expression data. It would be nice to help the reader by adding panels in the supplements showing normalized emax throughout to make it easier for the reader to see the impact of expression – now one can only see that expression isn't dramatically affected but too hard to do it one at a time by mutant.

To better see the effect of the GB2 “green” and “red” mutations, we have plotted the expression of the mutants versus the activation by GABA or by Rac-BHFF (agonist effect of the PAM in absence of GABA) in calcium release assays (see new Figure 5 – supplement 3, panel B and C).

Reviewer #2:GPCRs are major targets for drug development. A current promising avenue for GPCR drug development is to identify strategies to affect receptor signaling beyond simply targeting the endogenous ligand binding site. This would provide larger surface area to explore and better subtype specificity. Allosteric modulators do this by binding elsewhere in the receptor and enhance or attenuate effect of the orthosteric ligand. The general goal of this research was to investigate the mechanism of such effect in GABAB receptors. Finding the mechanism could inform development of future pharmaceutical drugs. GABAB receptors are members of class C GPCRs. This small family of GPCRs and is somewhat unique in that class C receptors form constitutive dimers. Recent structures of GABAB receptors from 4 different groups showed that 2 GABAB receptor PAMS (rac-BHFF and GS39783) bind at the dimeric interface which is unique among GPCRs. In this study, authors first verified this by performing functional assays and mutagenesis. They also showed the previously described constitutive activity of GABAB receptors and the direct agonist effect of these PAMs (called ago-PAM). Using mutation and structure-function approaches they showed the role of TM6 in both of these properties. Next, they authors tried to use similar assays to investigate the mechanism of agonist activity and modulation by these PAMs. Their results demonstrate the contributions of an internal cavity within the 7-transmembrane domain of GB2, dubbed "deep region", in the constitutive activity of GABAB receptor and the agonist effect of the PAMs but not the allosteric modulations.While these results provide an interesting twist to how allosteric modulators are perceived and that the pharmacological classification of compounds can be intertwined and not orthogonal, it provides few new mechanistic understanding. For example how does PAMs affect activation (does it modify the energy landscape? or induce a new conformation?) and is it possible to synthesize a PAM that is only a modulator and does not have direct agonist effect? Moreover, the methods section lacks practical details.

We agree with the reviewer. Our study clarifies how the current allosteric modulators of the GABAB receptor work. As other GABA-B ligands, the PAM are expected to modify the energy landscape of the receptor, since membrane receptors are highly dynamic protein, and both orthosteric and allosteric ligands are expected to modify, even slightly, its structural dynamics. Even though the most simple view is that GABAB PAMs favor active conformations stabilized by the orthosteric agonists, it is difficult to exclude that these PAMs induce a new conformation that would not be stabilized by the orthosteric ligands. This has been discussed in the revised part of the Discussion. Other studies would be necessary to address this issue.

The three GABAB PAMs investigated in this study have all an agonist activity, even though that of the GS39783 can be revealed only by the most sensitive functional assay (IP1 accumulation). Neural allosteric ligands (NAL) have been described for other GPCRs (Christopolous et al., (2014) Pharm Rev; Hellyer et al., (2018) Mol Pharm) indicating that allosteric modulators that do not have an agonist effect should be possible to obtain. To our knowledge, such compounds were not yet reported for the GABA-B receptor.

The method section (intracellular calcium assay and curve fitting) was better described in the revised version.

Specific comments:Line 110 – Is it possible the effect of GS39783 in the IP1 assay is due to nonspecific effect? Control titration experiments for figure 2B with cells transfected only with the chimeric G protein or the individual GABAB subunits + the chimeric G proteins would be highly relevant to this data.

We agree with the reviewer. We have now included IP1 data with chimeric G protein or the individual GABAB subunits + the chimeric G proteins for the GS39783 and the two other PAMs (see new Figure 2 – supplement 1).

Figure 3, figure supplement 1- While it is generally suggested that the ER retention signals inhibit trafficking of GABAB1 subunit to the surface, depending on expression level and cell type this could be leaky. Authors should provide GABAB1 only and GABAB2 only surface expression levels.

The results with the wild-type GABA_B1_ only and GABA_B2_ only have been shown in the revised Figure 3—figure supplement 1, panel A.

Figure 4, panel H. Previous work from authors and others (for example Koehl et al., 2019) has shown that for experiments on mGluR5, co-expression of neuronal excitatory amino acid transporter 3 (EAAT3) is necessary to be able to characterize the effect of modulators correctly. Without that, modulators can appear to show intrinsic agonist activity (for example Koehl et al., 2019). In the data in this panel, EAAT3 was not co-expressed.

It is a good remark from the reviewer. In Figure 5, panel H, the experiments have been performed in presence of the co-transfected glutamate transporter EAAT3 (also known as EAAC1), as well as in the cell surface quantification performed with the mGlu5 receptors and glutamate induced IP1 accumulation (Figure 5 —figure supplement 4E-F). It is now clearly written in this figure legend, in the revised text and in the Materials and methods section.

Authors have previously shown that GABAB receptors form extensive dimers of dimers and have characterized those. What is the effect of those higher order oligomers on the observed PAM effect?

It is a good question from the reviewer. Indeed, in our experimental conditions, we probably have a mix of heterodimers, dimer of dimers and higher order oligomers. Our lab recently showed that a strong negative effect between the GABA_B1_ binding sites exists within the GABAB oligomer, that limits G protein activation (Comps-Agrar et al., (2012) EMBO J; Stewart et al., (2018) Neuropharmacol). Accordingly, the effect of the PAM on signaling might be lower on the oligomers compared to the heterodimer. Experimental evidence would require to test the PAM effects on GABA_B1_ mutants (Comps-Agrar et al., (2012) EMBO J; Stewart et al., (2018) Neuropharmacol) that have less tendency to form dimer of dimers. But these experiments are difficult to perform since these mutants also impair the cell surface expression of the GABAB receptor, and we think they are outside the scope of the present study.

Line 166 – This explanation for the discrepancy is not clear. The different PAMs tested affect certain mutants in a distinct and sometimes opposite trends (potentiation versus reduction of IP1 accumulation). Specifically, Y810A, MYN-AA, M694A show opposing effects of the different PAMs, while the K792A mutant results in GS39783 showing the greatest potentiating effects.

We clarify the text in the revised version: “Of note, GS39783 is more sensitive to mutations than the other PAMs. It might be because the mutated residues are highly important for the binding of GS39783, or alternatively to its weakest agonist activity compared to Rac-BHFF and CGP7930 then resulting in a stronger loss of agonist activity of GS39783 on these mutants.”

Line 168 – Based on the data provided (figure 4E) and cited literature (Binet et., 2004; Sun et al., 2016), while the authors showed that CGP7930 rely on key residues in TM6 of GB2 to elicit its agonist effects, this does not imply that these residues are in contact with the compound, forming its binding site versus being a distal component of its allosteric network, or preclude the possibility that CGP7930 binds within or at a different external part of the TMD of GB2. Due to the lack of a structure of GABAB in complex with CGP7930 and the fact that past work shows CGP7930 can directly activate GB2 TMD, the assertion that CGP7930 directly binds at the TM6 interface is under-substantiated. In addition, mutations in the "deep region" within the GB2 TMD pocket, specifically mutants M14 and M15, in principle show that CGP7930 have ablated agonist activity – which can either be caused by the disruption of a "microswitch" necessary for CGP7930 agonism or disruption of a secondary binding site.

We agree with the reviewer. The possibility of another binding site for CGP7930 in GB2 TMD and the apparent controversy between the previous data reported by Binet et al., (2004) and our current data on CGP7930 need to be clarified. We think we did not reproduce the data reported in Binet et al., possibly due to the fact that the cells used in this previous study may have enough endogenous GB1 expression. Indeed, it was shown that the endogenous expression of GB1 in cell lines may varies, including in HEK293 cells (Xu et al., (2014) Angewandte Chemie). This reference has been added in the revised manuscript. Our current hypothesis is thus that the previous studies from Binet et al., (2004) and Sun et al., (2016) have proposed a wrong interpretation of the results, most probably due to endogenous expression of the GB1 subunit.

The reviewer suggests that the loss of agonist activity of CGP7930 on the GB2 mutants M14 and M15 could be due to the disruption of a binding site for this ago-PAM. Although we cannot totally exclude the possibility of an allosteric binding site of CGP7930 in the GB2 TMD, a similar loss of agonist activity were measured for the three PAMs on the same mutants (see new Figure 5supplement 3), while it is clear that GS39783 and rac-BHFF bind at the TM6 interface in the heterodimer as reported in the cryo-EM structures (Shaye et al., Nature, 2020; Mao et al., Cell Res 2020; Shen et al., Nature, 2021). Based on our mutagenesis analysis, we can thus reasonably propose that the main functional binding site for CGP7930 is most probably in the TM6s interface in the heterodimer.

Line 190- Kind of related to the previous comment, authors propose 2 general mechanisms for the ago-PAM effect of the compounds, essentially stabilization of the active state via binding of the PAM at the TM6 interface or within a second binding site. The type of experiments they present following up does not directly provide mechanistic insights to verify these models. The mutagenesis work generally reveals involvement of certain residues. This result is not very surprising as these compounds are allosteric and as it has been shown for example with deep mutational scanning on bAR, many mutations within the TMDs affect receptor signaling (Jones et al., eLife 2020).

We agree with the reviewer. Site-directed mutagenesis could induce long distance effects such as mutations in the TM6s interface, or in the GB2 TMD “deep region”, could impair a remote binding site in the GB2 TMD core. But it is interesting to observe that the three PAMs have similar effects on all the mutants in GB2 TMD, that could be explained mostly by a common binding site for the three PAMs in GB2 TMD. But due to the different chemical structures of the three PAMs, it is difficult to propose such a common binding site in GB2 TMD. A binding pocket that can accommodate these three PAMs is probably more possible at the TM6s interface.

Aside from his argument, these are not the only 2 possible mechanisms for the ago-PAM effect. One very plausible model is that the ago-PAM effect could be due to stabilization of an intermediate state, or even appearance of a new intermediate state upon PAM binding. These reasonable models are not discussed and not included in the mathematical modeling.

We agree with the reviewer, the ago-PAM effect could be due to the stabilization of an intermediate state of the receptor. We have now included this possibility in the revised version of the Discussion. But we think that including this intermediate state of the receptor in the mathematical model is beyond the scope of this study.

Methods section lacks important details for all the assays. For example, Line 397, what concentration of Fluo-4 AM was used? what is the assay buffer? what is the volume? what is the temperature that the assays were performed at?

In the revised version, the Materials and methods section for the intracellular calcium measurement was clarified as suggested by the reviewer.

“Intracellular ca^2+^ release in HEK293 cells was performed as previously described (Lecat-Guillet et al., 2017). Briefly, transfected cells in the 96 well plate were washed once with HBSS buffer (20 mM Hepes, 1 mM MgSO_4_, 3.3 mM Na_2_CO_3_, 1.3 mM CaCl_2_, 0.1% BSA and 2.5 mM probenecid) and pre-incubated with 1 μM ca^2+^-sensitive Fluo4 AM (Thermo Fisher Scientific) prepared in the HBSS buffer for 1 h at 37°C. Cells were washed once with HBSS buffer and 50 μL buffer was added into the wells before measuring in the multimode microplate reader (FlexStation 3, Molecular Devices). The fluorescence signals (excitation at 485 nm and emission at 525 nm) were then measured at intervals of 1.5 s for 60 s with the adding of 50 μL indicated compounds 20 s after the first reading by the reader automatically. The ca^2+^ response was given as the agonist-stimulated fluorescence increase, normalized according to the indication. Concentration response curves were fitted using “log(agonist) vs. response -- Variable slope (four parameters)” by GraphPad Prism software.”

Line 433 – Fitting equation, parameters, and constraints on the fits should be included. currently it is reported as "Activation- and Inhibition concentration response curves were fitted to the default equations of "log(agonist) vs. response -- Variable slope (four parameters)" and "log(inhibitor) vs. response -- Variable slope (four parameters)" within the software respectively."

To answer reviewer’s comment, the Materials and methods section was clarified:

“Activation concentration response curves were fitted to the default equation of “Log(agonist) vs. response – Variable slope (four parameters)” within the software, as Y=Bottom + (Top-Bottom)/(1+10^((LogEC50-X)*Hill slope)) with no constraints for the four parameters (X, log of concentration; Y, response; Top and Bottom, plateaus in same units as Y; LogEC50, same log units as X; and Hill slope, unit less) in the equation.”

Reviewer #3:Liu et al., investigate the mechanism of action of allosteric ligands that affect signaling at the GABAB receptor, a G protein-coupled receptor important in GABA function. The central question is how do these molecules achieve their unique activity, either as direct agonists or as allosteric potentiators of GABA. This study comes at the heels of several cryo-EM structures of GABAB receptors which have revealed the overall architecture of this heterodimeric receptor and the unique between transmembrane domain binding sites for two allosteric modulators that are the focus of this study (rac-BHFF and CGP7930). The authors perform a number of signaling studies to establish that two of these modulators have agonistic and allosteric activity, while one has only allosteric activity. Consistent with the binding site observed in previously determined structures, the authors determine that the transmembrane regions of the GABAB receptor are sufficient for the activity of agonistic positive allosteric modulators. Careful dissection of binding site residues suggests that the interactions made between the three allosteric molecules tested are slightly unique, though their binding sites are likely similar. The authors then test residues deep in the core of the seven transmembrane region of the GABAB subunits, and find that altering this region affects both constitutive activity and the agonistic properties of the modulators, while leaving their allosteric properties intact. With this, the authors propose that the deep site is crucially important for GABAB activation.While the conclusions specific to GABAB are supported by the authors data, some aspects need to be clarified or revised.1) It is unclear how well the authors model applies to GPCRs outside of the family C receptors. The authors suggest that the "deep" region is analogous to the sodium binding site of family A GPCRs. These regions differ substantially between the family A and family C GPCRs, and it is unlikely that they serve a similar structural rearrangement upon receptor activation. Furthermore, recent structures of family C receptors bound to G proteins suggest a very unique and divergent mode of G protein engagement compared to family A GPCRs. The authors' speculation around the similarity of activation mechanisms should take these considerations into account.

We agree with the reviewer. In the revised manuscript, we have strongly tuned down the generalizability of our discoveries outside the class C GPCRs. In the title, we have replaced “GPCR heterodimer” by “class C GPCR heterodimer”. In the abstract, we also removed the idea that the “deep region” reported here in GB2 is functionally conserved in class A GPCR.

2) The authors suggest that their findings regarding GABAB modulators that bind at interface between two transmembrane domains likely has implications outside of the family C receptors. The authors should provide appropriate context that, to date, no family A GPCRs have conclusively been shown to form constitutive dimers similar to GABAB or other family C GPCRs.

We agree with the reviewer. In the revised manuscript, we have strongly tuned down the generalizability of our discoveries outside the class C GPCRs. At the end of the Discussion, we just mention that ligands acting at a dimer interface may potentially be interesting tools for other GPCRs, even if they generally form for transient dimers.

3) A central new observation in this study is that the deep site is important for constitutive activity and agonistic allosteric modulator activity. The authors propose that this site is a hub for the activation mechanism. Another potential explanation, however, is that the mutations simply stabilize the inactive conformation of the GABAB compared to the wild-type receptor, which has a combined affect on both constitutive activity and the relatively weak agonistic activity of allosteric modulators like rac-BHFF. Indeed, Figure5-supplement 2B shows a rightward shift for GABA dose response for these mutants. The authors should more explicitly discuss potential caveats to their interpretation of the signaling data.

It is a very good remark from the reviewer. We could not exclude that the mutations in the “deep region” of GB2 TMD stabilize the inactive conformation of GABA-B (this possibility is now considered in the revised Discussion). As proposed by the reviewer, it could explain the rightward shift of GABA dose response for these mutants (see new Figure 5-supplement 3). But this hypothesis is difficult to test experimentally. In the revised text, we better discussed the potential caveats to the interpretation of the signaling data.

I recommend that the authors revise their speculation that the observations made in this work have general implications for the broader GPCR family outside of family C GPCRs.

We agree with the reviewer. This was done to make the manuscript more attractive to the editor. In the revised version, we have strongly tuned down the generalizability of our discoveries outside the class C GPCRs, including in the title where we have replaced “GPCR heterodimer” by “class C GPCR heterodimer”. We also removed the idea that the “deep region” observed here in GB2 is functionally conserved in class A GPCR. In the Discussion on class C GPCRs, and we just mention that ligands acting at a dimer interface may potentially be interesting tools for other GPCRs, even if they generally form for transient dimers.